# DYNAMIC MIXTURE-OF-EXPERTS FOR INCREMENTAL GRAPH LEARNING

## ABSTRACT

Graph incremental learning is a learning paradigm that aims to adapt models trained on previous data to continuously incremented data or tasks over time without the need for retraining on the full dataset. However, regular graph machine learning methods suffer from catastrophic forgetting when applied to incremental learning settings, where previously learned knowledge is overridden by new knowledge. Previous approaches have tried to address this by treating the previously trained model as an inseparable unit and using regularization, experience replay, and parameter isolation to maintain old behaviors while learning new knowledge. These approaches, however, do not account for the fact that not all previously acquired knowledge is equally beneficial for learning new tasks, and maintaining all previous knowledge and the latest knowledge in a single model is ineffective. Some prior patterns can be transferred to help learn new data, while others may deviate from the new data distribution and be detrimental. To address this, we propose a dynamic mixture-of-experts (DyMoE) approach for incremental learning. Specifically, a DyMoE GNN layer adds new expert networks specialized in modeling the incoming data blocks. We design a customized regularization loss that utilizes data sequence information so existing experts can maintain their ability to solve old tasks while helping the new expert learn the new data effectively. As the number of data blocks grows over time, the computational cost of the full mixture-of-experts (MoE) model increases. To address this, we introduce a sparse MoE approach, where only the top-$k$ most relevant experts make predictions, significantly reducing the computation time. Our model achieved 5.47% relative accuracy increase compared to the best baselines on class incremental learning with minimal computation increase, showing the model's exceptional power.

## 1 INTRODUCTION

Graph neural networks (GNN) achieved great success in modeling graph data and have many applications, such as recommender systems (Wang et al., 2021), drug discovery (Gaudelet et al., 2021), and traffic forecasting (Jiang & Luo, 2022). However, in many real-world settings, the graph is dynamic, starting small and expanding over time, and the training data arrive as sequences of data blocks. Naive approaches train on the full graph whenever new data appears, which incurs expensive computational costs due to repetitive training on old data. On the other hand, simply finetuning conventional GNNs on the new data leads to catastrophic forgetting, where the model's prediction shifts toward the new data distribution and forgets how to handle previously learned tasks upon encountering new data (Zhang et al., 2023b; 2024; Cui et al., 2023; Xu et al., 2020). This motivated a series of *continual learning* research to tackle this problem (Yuan et al., 2023; Febrinanto et al., 2023; Wu et al., 2024).

Pioneering efforts focused on adapting incremental learning approaches for other data modalities to the graph domain (Zhou & Cao, 2021; Xu et al., 2020; Sun et al., 2023). However, they ignore the fact that nodes and edges are not independent and identically distributed (i.i.d.) in the graph learning scenario (Wang et al., 2022; 2020). In the vision and language domain, individual image or text data points do not affect each other, and future data blocks do not impact the data distribution of the existing data blocks. In contrast, new graph data blocks connect to existing data via edges and could significantly change existing data distribution. For example, an incoming data block can add edges between two disconnected components in an existing graph, drastically changing the graph topology

Figure 1: Left: Data blocks arrive in sequence. Right: Different connection types of three data blocks. Our proposed method activates dedicated experts when inferring relevant data blocks.

and, subsequently, the learned model behavior. Incremental blocks in the graph domain break the i.i.d assumption of data in most incremental learning approaches from the vision and language domains. It makes graph incremental learning an even more challenging scenario than incremental learning in other domains.

Subsequent efforts tackled the problem in several ways (Tan et al., 2022; Xu et al., 2020; Wang et al., 2022). For instance, PI-GNN (Zhang et al., 2023a) rectified the old model on the graph modified by the new data. TWP (Liu et al., 2021) identified topology-aware parameters to stabilize the model under graph structure shift. DiCGR (Kou et al., 2020) breaks relation triplets to components to better capture graph structures.

These methods show improvements in the graph setting compared to the naive adaptation of incremental learning methods from other domains. However, a commonality of these approaches is that they build the new model upon an inseparable old model. Specifically, Elastic Weight Consolidation (EWC) (Kirkpatrick et al., 2017) used the old model parameters as the single regularization target for all parameters; Experience Replay (ER) (Zhou & Cao, 2021) trained the model using all saved subsets of nodes from old data blocks; Parameter Isolation (PI-GNN) (Zhang et al., 2023a) froze all old model parameters and used an additional network to modify the model output.

While these methods effectively keep the old patterns, they assume all past data blocks have the same impact when learning new patterns, ignoring different correlations among them. For example, in Figure 1, blue, red, and green nodes represent data blocks one, two, and three that arrive in order in all three cases, and we update the model whenever a data block arrives. Block one and two are identical in all cases, while block three is isolated, connected to block two, and connected to blocks one and two in cases A, B, and C, respectively. Existing approaches will modify the knowledge learned from blocks one and two in case A to accommodate new knowledges in block three regardless of the connection type, causing forgetting. However, in case A, the blocks are entirely isolated, and the third block can be independently learned without modifying the model and parameters learned from the other two blocks. Existing approaches would still update the entire model, ignoring the factor that, in case A, each data block can be learned independently without forgetting. Likewise, in case C, the third block only needs information from block one, but existing approaches would still apply knowledge obtained from block two, which leads to both negative transfer and forgetting.

To tackle this problem, we propose a Dynamic Mixture-of-Expert (DyMoE) module to use separate expert networks to model different data blocks with a gating mechanism to synthesize information from the most relevant experts. Specifically, the module has the same number of experts as trained data blocks, and each expert has a corresponding gating vector. Experts are dedicated to learning from their corresponding data blocks. Unlike existing works that process all previously learned knowledge equally, given input, our module first computes the similarity between the input and each gating vector to determine the relevance of experts to the input, then calculates the expert outputs for the input, and finally uses the relevance to combine the outputs with a weighted sum. This approach explicitly considers the correlation between different experts and data blocks. For the same example in Figure 1, we train three separate experts with specialization in their corresponding data blocks. We then compute the relevance of the experts to the input. The experts with higher relevance have a higher impact on the prediction. This approach dynamically adjusts the combination of knowledge

from different data blocks; less impactful experts are disabled during inference to reduce misleading information. When a new data block arrives, we append a new expert dedicated to the new data block without interfering with the knowledge of existing experts during training. To ensure each expert focuses on the assigned data block, we propose a block-guided loss as a training objective that enforces a high relevance score of experts to the input from their corresponding data blocks, greatly reducing catastrophic forgetting while allowing flexible querying of old knowledge.

As the receptive field of each graph neural network layer may change every time a new data block is merged into the original graph, we organically fuse the DyMoE module into each GNN layer to handle the unique data dependency challenge in the continual graph learning domain. Specifically, we interleave the DyMoE module into each layer so the model knows neighbor nodes from different data blocks and encodes them differently to learn data block specialized message passing. Moreover, we propose a sparse variant, inspired by Shazeer et al. (2017), that only considers the most relevant experts to reduce the computational cost incurred by additional experts, significantly reducing the computation complexity while maintaining high accuracy. In this paper, we

- Identified the issue of existing continual learning methods that ignore the correlation between different data blocks.
- Designed a DyMoE module with specialized experts for each data block and proposed data block-guided loss to minimize the negative interference between experts.
- Interleave the DyMoE module into GNNs to address the data shift problem unique to graph continual learning.
- Developed a sparse version of the DyMoE module so the model is both efficient and effective.

In our empirical evaluation, our results show up to 10% and on average 5.47% relative accuracy improvement over the best baseline on class incremental learning setting. The model also demonstrates strong results in instance incremental settings. We also show that our approach can achieve close results to the upper-bound retraining method using significantly less time for training, further validating the model's efficacy.

## 2 PRELIMINARIES

**Graph Incremental Learning.** This paper focuses on incremental learning for node classification. Specifically, we follow the widely adopted problem formulation (Yuan et al., 2023; Febrinanto et al., 2023), and aim to incrementally learn from a graph data block sequence $D = \{G_1, ..., G_t\}$, and each data block is a graph $G_i = (V_i, E_i, Y_i)$ where $V_i$ is the set of nodes, and $E_i$ is the set of edges, and $Y_i$ is the classification labels of the nodes. Future graph snapshots expand on existing graphs, and $G_i$ is a subgraph of $G_j$ for $i < j$. We additionally use $\Delta G_i = (V_i \setminus V_{i-1}, E_i \setminus E_{i-1})$ to represent the graph delta between $G_i$ and $G_{i-1}$. We use $b(v)$ to indicate the index of the data block where the node $v$ first appears. In the incremental learning setting, data arrive in order, and the $i$-th model is only trained and evaluated on $(G_1, ..., G_i)$ without any knowledge about future graphs. The goal is to maximize the overall accuracy on each data block while minimizing the performance drop on previous data blocks. If the classes in $Y_i$ persist throughout all blocks, we refer to the task as instance-incremental learning (Van de Ven et al., 2022). If the classes in $Y_i$ are disjoint, we refer to the task as class-incremental, where new data blocks also bring in new classes (Zhang et al., 2022), and the model needs to classify a sample without knowing its corresponding block during inference.

The naive solution is to train a model on the full graph $G_i$ for every block. However, this requires retraining on all old data multiple times, incurring huge computational costs. Incremental learning methods aim to train only on the graph delta while maintaining good performance on the old data.

To evaluate a model, let $a_{i,j}$ be the accuracy of all evaluation nodes in $G_i$, evaluated by the model after training $G_j$, which is a superset of evaluation nodes in $G_i$ and $i \leq j$. We evaluate the overall model performance by Average Accuracy (AA) and Average Forgetting (AF),

$$AA = \frac{1}{t} \sum_{i=1}^{t} a_{i,i}, \quad AF = \frac{1}{t} \sum_{j=1}^{t} \frac{1}{j} \sum_{i=1}^{j} a_{i,j} - a_{i,i} \tag{1}$$

Figure 2: Pipeline of DyMoE GNN. Left: Each GNN layer has a message-passing module and a DyMoE module. We compute gating values from the node representations and the gating vectors. During training, we compute a block-guided loss between the gating values and the data block index for correct expert selection. Right: When a new data block arrives, we add a new expert and a gating vector to the DyMoE module. In the sparse case, only the most important experts are used.

where $t$ is the number of data blocks. AA evaluates the model's average accuracy right after the model is trained on a data block, while AF evaluates the model's ability to retain knowledge from previous data blocks. The goal of an incremental learning method is to maximize AA and minimize AF.

**Graph Neural Networks** Graph neural networks iteratively update a node's embeddings from their neighbor nodes through message-passing layers (Gilmer et al., 2017). Specifically, for a graph $G = (V, E)$, the $i$-th layer of a $T$-layer GNN is,

$$\boldsymbol{h}_v^{(i+1)} = COMB(\boldsymbol{h}_v^{(i)}, AGGR(\{\boldsymbol{h}_u^{(i)}|u \in \mathcal{N}(v)\})), \quad v \in V, \quad \mathcal{N}(v) = \{u|(v,u) \in E\} \quad (2)$$

where $\mathcal{N}(v)$ are the direct neighbors of $v$. Different GNN designs differ mainly by the combine (COMB) and aggregate (AGGR) functions.

## 3 DYNAMIC MIXTURE-OF-EXPERTS GRAPH NEURAL NETWORK

This section first introduces the Dynamic Mixture-of-Experts (DyMoE) module that dynamically increases the number of experts for new data blocks. We then describe the integration of DyMoE and GNN for effective graph incremental learning. To overcome the efficiency issue with long data sequences, we propose Sparse DyMoE to reduce the complexity of our framework. The overall architecture of the framework is shown in Figure 2.

### 3.1 DYNAMIC MIXTURE-OF-EXPERTS MODULE

Conventional mixture-of-experts (MoE) models create networks of the same architecture and apply a gating mechanism to combine the networks' outputs using a weighted sum (Shazeer et al., 2017). The number of experts is fixed after initialization. However, to accommodate new data blocks, MoE models suffer from the same issue as in other continual learning methods. They still need to adjust the weights of all previous experts, leading to forgetting. To mitigate this, we propose the DyMoE module, adding one expert for every new data block without modifying previously trained experts. Let $\mathcal{F}$ be a class of neural networks with the same architecture, and $f_\theta \in \mathcal{F}$ be an instance of the network parametrized by $\theta$. Specifically,

$$\boldsymbol{h} = f_\theta(\boldsymbol{x}) \quad \boldsymbol{x} \in \mathcal{R}^n, \boldsymbol{h} \in \mathcal{R}^m, f_\theta \in \mathcal{F} \quad (3)$$

where $\boldsymbol{x}$ and $\boldsymbol{h}$ are the input and output to the network, and $n$ and $m$ are the input and output dimensions. Given an incremental data sequence $D = \{(X^{(1)}, Y^{(1)}), ..., (X^{(k)}, Y^{(k)})\}$, DyMoE handles the first data block like a conventional neural network. Specifically, it minimizes the empirical

loss,

$$\arg\min_{\theta_1} \frac{1}{|X|} \sum_i^{|X|} \mathcal{L}(y_i, f_{\theta_1}(\boldsymbol{x}_i)) \tag{4}$$

The loss function $\mathcal{L}$ is task dependent, and we use cross-entropy loss for classification. For the second data block, we will add one expert and gating vectors to the overall model. To compute the output, we have

$$\boldsymbol{h} = f_{\{\theta_1,\theta_2\}}(x) = \alpha_1 f_{\theta_1}(x) + \alpha_2 f_{\theta_2}(x), \quad \alpha_i = \frac{exp(s(\boldsymbol{x},\boldsymbol{g}_i))}{exp(s(\boldsymbol{x},\boldsymbol{g}_1)) + exp(s(\boldsymbol{x},\boldsymbol{g}_2))} \quad i \in \{1,2\} \tag{5}$$

where $\boldsymbol{g}$ are gating vectors associated with each expert, $s(\cdot,\cdot)$ is a similarity measure, and we use softmax on the similarities to compute the importance of each expert for the input. Note that this formulation is the same as existing MoE approaches, and the key difference is that the number of experts dynamically increases as more data arrive. Subsequent data blocks follow the same procedure, where the output is computed as,

$$\boldsymbol{h} = f_{\{\theta_1,...,\theta_t\}}(x) = \sum_{i=1}^t \alpha_i f_{\theta_i}(\boldsymbol{x}), \quad \alpha_i = \frac{exp(s(\boldsymbol{x},\boldsymbol{g}_i))}{\sum_{j=1}^t exp(s(\boldsymbol{x},\boldsymbol{g}_j))} \tag{6}$$

When training on a new data block $t$, we only optimize the new expert and its corresponding gating vector, specifically,

$$\arg\min_{\theta_t,\boldsymbol{g}_t} \mathcal{L}_{cls}, \mathcal{L}_{cls} = \frac{1}{|X_t|} \sum_i^{|X_t|} \mathcal{L}(y_i, f_{\{\theta_1,...,\theta_t\}}(\boldsymbol{x}_i)) \tag{7}$$

Intuitively, this training scheme completely preserves the knowledge obtained from previous data blocks. Ideally, when the gating vectors are perfectly trained to distinguish which data block a particular data point belongs to, the model can **fully recover** the output of that data point, eliminating forgetting. While the gating vectors are trained simultaneously with the experts, the first is not trained because applying softmax to a single value results in a trivial weight (value one). Hence, we propose using the input's mean to initialize the first gating vector. Specifically,

$$\boldsymbol{g}_1 = \frac{1}{|X_1|} \sum_{i=1}^{|X_1|} \boldsymbol{x}_i \tag{8}$$

as it minimizes the sum of $l_2$ distance between $\boldsymbol{g}_1$ and $X$. Setting the first gating vector to the empirical mean ensures that a data block has high gating values if it belongs to the first data block without direct training.

While the experts can preserve learned knowledge, the new experts are randomly initialized and start with trivial predictions on all data. The model will rely on the existing trained experts to make predictions, though they may carry old, potentially suboptimal, knowledge regarding the new data block. The gating vectors, including the new one, will tend to select the old experts during training. The model will be trapped at the local minimum without properly training the new dedicated experts. Figure 6 shows that direct training will not result in specialized experts. Hence, we need to inject the information about the correct experts for our dynamically initialized new modules. This is difficult in conventional MoE because of the lack of supervision for correct experts. However, in continual learning, data arrive in blocks, and since experts are designed to handle individual data blocks, we know exactly which expert a particular training data point should be assigned to. We propose a **block-guided regularization** to train the gating vectors for correct expert assignment. Specifically, for an arbitrary data point $\boldsymbol{x}$, in addition to its classification loss, we add a cross-entropy loss between the gating values of all experts and the data point's corresponding data block index $b(\boldsymbol{x})$. The computation is valid because the number of experts equals the number of witnessed data blocks. The loss forces an expert's corresponding data and gating vector to have large similarities, maximizing the likelihood of using the correct expert to generate output for the data. Specifically,

$$\mathcal{L}_{BL} = CE(Softmax(s(\boldsymbol{x},\boldsymbol{g}_1),...,s(\boldsymbol{x},\boldsymbol{g}_t)), OneHot(b(\boldsymbol{x}),t)), \boldsymbol{x} \in P \tag{9}$$

where $CE$ is cross-entropy loss, $OneHot(j,t)$ generates a $t$-dimensional one-hot vector whose $j$-th entry is one, and $P$ is the training set in the current data block. Note that if we naively take $P$ as the

new samples in the most recent data block $X_t$, all of them will have the same data block index (the last index), causing the model to always use the last expert. Hence, we store a small sample set from each data block as memory set, $M_i \subset X_i$ and $|M_i| \ll |X_i|$, and take $P = \bigcup_i^{(t-1)} M_i \cup X_t$, so the model can adjust the gating values accordingly.

Note that we only use such information during training, and the model does not need the time information, or the data block that a data point belongs to, during inference, making the model perfectly viable for difficult tasks such as class-incremental learning. The overall training loss is,

$$\mathcal{L} = \mathcal{L}_{cls} + \beta \mathcal{L}_{BL} \tag{10}$$

where $\beta$ is a hyperparameter controlling the strength of regularization. The combined framework essentially attempts to train a data-block-dedicated classifier and out-of-distribution detectors for every data block. The gating mechanism gives high weight to in-distribution experts while minimizing the impact of out-distribution experts. While this approach applies to arbitrary data modality, it is particularly critical in the graph learning setting, where a node's neighbor might be from different data blocks and require different processing. We elaborate more on this in Section 3.2. We theoretically show the advantages of our proposed model over the Parameter Isolation (PI) (Zhang et al., 2023a) approach, a representative architectural approach for continual learning.

**Theorem 1.** *For an arbitrary continual learning problem, suppose a PI model obtains a cross-entropy loss $\mathcal{L}_{PI}$, there exists a parametrization of DyMoE that achieves cross-entropy loss $\mathcal{L}_{Dy} = \mathcal{L}_{PI}$. When the data sequence follow a mixture of Gaussian distribution, we have $\mathcal{L}_{Dy} \leq \mathcal{L}_{PI}$.*

The proof is in Appendix A. In the proof, we first show that DyMoE is at least as powerful as PI. We then show under the Gaussian Mixture assumption of the input data block sequence; the DyMoE obtains strictly lower loss, which shows the model's superiority.

In practice, the memory set is very small to ensure efficiency, but we jointly train on it with the full dataset from the new data block, which can give the model a biased understanding of the data distribution (i.e. most of the data are from the last data block). Hence, we propose a *data balancing* training procedure, where, after the regular training epochs, we collect the memory set for the new data block, combine it with all previous training memory sets, and train a few epochs on them to reflect the actual distribution of the entire input sequence. Because the memory sets are very small subsets, they bring minimal computation costs.

## 3.2 DYNAMIC MIXTURE-OF-EXPERT GRAPH NEURAL NETWORK

We then introduce fusing the DyMoE with a graph neural network. Note that the DyMoE module does not assume any specific network architecture, and a naive solution can treat a multi-layer GNN as $\mathcal{F}$. However, this ignores the unique property of graph data in continual learning, where new data can change the existing graph's overall topology and the representation learned for the old data. For example, in Figure 3, the target node is from data block one but is later connected to nodes in blocks two and three. However, as shown in the computation graph of the naive approach, it will still use expert one to process neighbor nodes of the target node from blocks two and three, while expert one does not know the new data blocks. The experts are completely isolated, and we cannot use future expert information to correct the misrepresentation of the

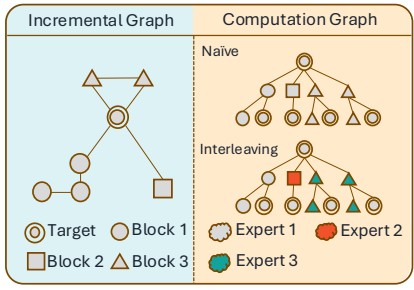

Figure 3: Comparison between computation graphs of two approaches.

neighbor node. Consequently, the representation of the target node is considerably compromised. Essentially, compared to traditional incremental learning scenarios, where the old model still performs well on old data, the modified topology in the graph causes the old model to shift from its original prediction and cause a performance decrease. Hence, we propose interleaving the DyMoE modules into each GNN layer to correct such a shift. Specifically,

$$\boldsymbol{h}_v^{(i)} = f_{\{\theta_1,\ldots,\theta_t\}}(\boldsymbol{h}_v^{(i-1)} + \epsilon \cdot AGG(\{\boldsymbol{h}_u^{(i-1)}|u \in \mathcal{N}(v)\}) \tag{11}$$

we use DyMoE as the COMB function and instantiate each expert as an MLP, and the overall design is a natural extension of a succinct message-passing network as in GIN (Xu et al., 2018). However, the

interleaving DyMoE design can be easily extended to other GNN architectures. The key difference between this and naive approaches is that we combine information from all experts after each message passing layer but not from the final output of multiple GNN layers. Take Figure 3 as an example, for the same target node, the proposed approach allows the output to absorb information from future experts and correct the features learned in previous data blocks to adapt to the new graph context rather than combining the compromised node representation at the end of all GNN processing.

Finally, we need to accommodate the block-guided regularization loss to a more fine-grained version for the interleaving design. Instead of using the target node's corresponding data block as the regularization target, we use each neighbor node's own corresponding data block as the target. Specifically,

$$\mathcal{L}_{BL,GNN} = \sum_{i=1}^{T} \sum_{v \in V} \mathcal{L}_{BL}(\boldsymbol{h}_v^{(i)}, b(v)) \tag{12}$$

where $b(v)$ is the corresponding block index of node $v$. Intuitively, when the neighbor and the target nodes are from different data blocks, we still want the most relevant expert to be of higher importance than the expert corresponding to the data block of the target nodes. Let's take Figure 3 as an example again. For the final target node, we expect the experts used to compute the intermediate representation of its neighbors to be Expert 1, Expert 2, Expert 3, and Expert 3 from left to right. Compared to applying block-guided loss in other modalities, this additionally addresses the topological and context shift problem in the graph learning domain by ensuring the corrected representation of neighbor nodes from different data blocks.

### 3.3 Sparse Dynamic Mixture-of-Experts GNN

While the proposed DyMoE GNN allows effective knowledge preservation and updates specialized for graph data, it incurs additional computation cost for the dynamically increasing experts. With more data blocks, we can have too many experts whose computational burden overwhelms the performance benefits of the module. Inspired by previous works on Sparse MoE (Shazeer et al., 2017), we introduce sparsity into the system to improve its efficiency. To that end, we modify Equation 6 so that only the experts with the top-k importance score are used to generate predictions. Specifically,

$$\boldsymbol{h} = \sum_{i=1}^{t} \alpha f_{\theta_i}(\boldsymbol{x}), \quad \alpha_i = Softmax(TopK(s(\boldsymbol{x}, \boldsymbol{g}_j))) \tag{13}$$

Because we only use the top-k most essential experts, we do not need to propagate gradients and compute the output of each expert, which significantly reduces the training and inference cost.

Since the last expert and gating are randomly initialized, the model may ignore them because they produce meaningless predictions at the beginning. To mitigate this, we follow Sparse MoE (Shazeer et al., 2017) to tweak the gating values during training randomly so all experts have similar selection chances, and the new experts and gates can gradually learn to correctly predict the new data block.

## 4 Related Work

**Incremental Learning** is extensively explored in the deep learning literature, including computer vision (Kirkpatrick et al., 2017; Li & Hoiem, 2018; Lopez-Paz & Ranzato, 2017) and natural language processing (Ke & Liu, 2022; Sun et al., 2020; Mi et al., 2020). The approaches can be roughly divided into three categories: **Regularization-based** methods constrain the deviation of the new model from the trained model to retain knowledge (Kirkpatrick et al., 2017; Zenke et al., 2017; Aljundi et al., 2018); **Experience-Replay** approaches add a small subset of previous data blocks to the current training set as a way to maintain previous knowledge (Lopez-Paz & Ranzato, 2017; Rolnick et al., 2019; Chaudhry et al., 2021); **Architectural** approaches maintain learned knowledge via assigning model parameters to specific data (Aljundi et al., 2017; Ebrahimi et al., 2020; Li & Hoiem, 2018). Our method falls into the architectural category. Some existing work also considers separate modules for each data block Aljundi et al. (2017); Rusu et al. (2016), but they focus on the task-incremental scenario, while our method handles both that, and the more challenging class-incremental case. More importantly, they do not account for structural shift in graph incremental learning, whereas our approach handles this well.

Table 1: Average accuracy and average forget of class incremental datasets.

| | CoraFull | | Reddit | | Arxiv | | DBLP | |
|---|---|---|---|---|---|---|---|---|
| | AA | AF | AA | AF | AA | AF | AA | AF |
| Pretrain | $17.51_{\pm3.51}$ | $0.00_{\pm0.00}$ | $35.99_{\pm2.93}$ | $0.00_{\pm0.00}$ | $27.86_{\pm3.76}$ | $0.00_{\pm0.00}$ | $49.40_{\pm2.17}$ | $0.00_{\pm0.00}$ |
| Online | $38.27_{\pm4.20}$ | $-23.51_{\pm2.61}$ | $28.94_{\pm0.12}$ | $-33.73_{\pm0.13}$ | $38.96_{\pm4.45}$ | $-36.84_{\pm3.73}$ | $47.29_{\pm4.37}$ | $-18.18_{\pm3.87}$ |
| EWC | $39.14_{\pm3.42}$ | $-22.68_{\pm3.42}$ | $31.16_{\pm2.85}$ | $-31.98_{\pm2.67}$ | $42.08_{\pm3.96}$ | $-30.75_{\pm2.35}$ | $50.19_{\pm1.82}$ | $-19.55_{\pm2.01}$ |
| LWF | $43.01_{\pm3.76}$ | $-18.40_{\pm3.20}$ | $47.74_{\pm3.03}$ | $-27.80_{\pm3.15}$ | $40.01_{\pm2.65}$ | $-30.72_{\pm3.31}$ | $53.15_{\pm3.30}$ | $-15.28_{\pm2.91}$ |
| ER-GNN | $71.08_{\pm0.23}$ | $-10.95_{\pm0.24}$ | $81.35_{\pm2.39}$ | $-8.71_{\pm0.73}$ | $57.09_{\pm2.21}$ | $-23.65_{\pm2.10}$ | $55.58_{\pm2.26}$ | $-9.59_{\pm1.89}$ |
| PI-GNN | $68.27_{\pm2.04}$ | $-8.79_{\pm0.50}$ | $84.13_{\pm1.43}$ | $-6.57_{\pm0.88}$ | $58.46_{\pm1.63}$ | $-16.00_{\pm1.09}$ | $\mathbf{59.18}_{\pm3.03}$ | $-11.12_{\pm2.07}$ |
| C-GNN | $78.90_{\pm1.13}$ | $-8.27_{\pm0.82}$ | $86.75_{\pm2.13}$ | $-6.06_{\pm0.47}$ | $63.65_{\pm1.95}$ | $-14.19_{\pm3.03}$ | $57.81_{\pm2.24}$ | $-9.79_{\pm1.28}$ |
| DyMoE | $\underline{80.97}_{\pm0.58}$ | $\mathbf{-5.22}_{\pm0.49}$ | $\mathbf{93.28}_{\pm0.19}$ | $\mathbf{-2.98}_{\pm0.36}$ | $\mathbf{68.06}_{\pm1.54}$ | $\underline{-10.68}_{\pm1.97}$ | $\underline{57.85}_{\pm3.17}$ | $\mathbf{-7.42}_{\pm1.93}$ |
| DyMoE (k=3) | $\mathbf{81.33}_{\pm0.85}$ | $\underline{-5.69}_{\pm1.12}$ | $\underline{91.57}_{\pm0.58}$ | $\underline{-3.46}_{\pm0.39}$ | $\underline{67.25}_{\pm0.97}$ | $\mathbf{-9.54}_{\pm0.69}$ | $57.75_{\pm2.96}$ | $\underline{-7.51}_{\pm1.82}$ |
| Retrain | $79.97_{\pm0.29}$ | $-4.63_{\pm0.53}$ | $96.51_{\pm0.13}$ | $-1.12_{\pm0.03}$ | $80.16_{\pm1.96}$ | $-6.33_{\pm1.06}$ | $67.54_{\pm2.02}$ | $-2.02_{\pm0.83}$ |

**Graph Incremental Learning.** Different from i.i.d. data, graph data suffer from distribution shifts in the incremental learning setting. To overcome this novel challenge, architectural approaches including, PI-GNN (Zhang et al., 2023a), FGN (Wang et al., 2022), and HPN (Zhang et al., 2023b), use newly initialized model components to learn new knowledge. Experience replay approaches like DyGRAIN (Kim et al., 2022), ER-GNN (Zhou & Cao, 2021), and Continual GNN (Wang et al., 2020) explicitly retrains old nodes selected from graph-related criterion. Regularization approaches such as TWP (Liu et al., 2021), GraphSail (Xu et al., 2020), and GPIL (Tan et al., 2022) identify and minimize a regularization loss to mediate structural shift and correct predictions. However, because these models treat old models as inseparable units, they ignore different interaction types between data blocks. Meanwhile, our experts are dedicated to individual data blocks, facilitating conditional adaptation to new data.

## 5 EXPERIMENTS

We aim to answer the following research questions in the experimental evaluation: **Q1**: Does the proposed DyMoE framework achieve good empirical performance while maintaining good efficiency? **Q2**: How does the memory size impact the performance of the model? **Q3**: The framework has several components, how does each component impact its behavior? **Q4**: Does our training strategy actually encourage dedicated experts? Implementation details and data descriptions can be found in Appedix C.

### 5.1 QUANTITATIVE RESULTS

To answer **Q1**, we evaluate the model performance with average accuracy (AA) and average forget (AF) on class incremental datasets (CoraFull (Weber et al., 2019), Reddit (Hamilton et al., 2017), Arxiv (Hu et al., 2021), DBLP-small (Tang et al., 2008)), and data incremental datasets (Paper100M(Hu et al., 2021), Elliptic (Weber et al., 2019), Arxiv, DBLP-small). We compared experience-replay baselines (ER-GNN (Zhou & Cao, 2021), continual-GNN (C-GNN) (Wang et al., 2020)), architectural baselines (LWF (Li & Hoiem, 2018), PI-GNN (Zhang et al., 2023a)), and regularization baselines (EWC (Kirkpatrick et al., 2017)). We also compared with the pretrain baseline, where we only train the model on the first date block and infer all future data blocks; the online baseline, where we directly fine-tune the old model with new data blocks; and the retrain baseline, where we retrain on all data blocks whenever new data blocks arrive. We provide the results of our dense model without the sparse DyMoE module and the sparse version with $k = 3$.

We show the experiment results of class incremental setting in Table 1. From the results, we can see our method significantly improves over existing baselines for both AA and AF. We reach an average of 5.47% improvement in AA and 34.64% reduction in AF. The solid empirical results showed the superiority of the DyMoE design and validated our theory. Notably, our framework even achieved AA better than retraining on the CoraFull dataset, showing that separate experts enable better knowledge transfer between different data blocks. Comparing the dense and sparse DyMoE,

Table 2: Average accuracy and average forget of instance incremental datasets.

| | Paper100M | | Elliptic | | Arxiv | | DBLP | |
|---|---|---|---|---|---|---|---|---|
| | AA | AF | AA | AF | AA | AF | AA | AF |
| Pretrain | 59.81±2.41 | 0.00±0.00 | 91.69±3.81 | 0.00±0.00 | 62.81±1.82 | 0.00±0.00 | 57.03±2.65 | 0.00±0.00 |
| Online | 65.10±2.51 | -3.57±0.83 | 94.97±0.23 | -0.89±0.14 | 70.05±0.91 | -1.11±0.03 | 66.52±1.58 | -3.90±0.57 |
| EWC | 72.86±1.95 | -3.19±1.42 | 94.89±0.16 | -0.90±0.01 | 70.03±0.54 | -1.30±0.18 | 66.25±1.46 | **-2.38**±0.79 |
| LWF | 70.07±1.88 | -3.84±1.67 | 92.78±0.42 | 0.73±0.11 | 68.47±0.61 | -1.34±0.06 | 67.77±2.03 | -2.93±0.56 |
| ER-GNN | 81.46±1.85 | -3.67±0.64 | 96.80±0.24 | 0.01±0.01 | 69.98±0.07 | -1.18±0.05 | 67.48±2.39 | -3.12±0.67 |
| PI-GNN | 82.53±1.37 | -4.18±1.29 | 93.44±0.28 | **0.87**±0.09 | **71.59**±0.13 | -1.63±0.85 | 66.12±2.18 | -3.99±1.02 |
| C-GNN | 81.34±1.08 | -4.58±1.56 | 96.05±1.16 | -1.06±0.30 | 70.78±1.08 | -1.35±0.36 | 66.96±1.16 | -3.71±1.24 |
| DyMoE | **83.97**±1.16 | **-2.37**±0.83 | 96.30±0.01 | -0.20±0.09 | 69.81±0.17 | **-0.81**±0.09 | 67.51±0.47 | -3.01±0.84 |
| DyMoE (k=3) | 82.93±1.53 | -3.31±1.58 | **97.01**±0.32 | -0.40±0.07 | 69.05±0.82 | -1.21±0.47 | **67.79**±0.38 | -3.46±1.37 |
| Retrain | 86.15±0.49 | -0.35±0.04 | 98.13±0.03 | 0.14±0.02 | 73.01±0.10 | 0.34±0.29 | 68.59±1.27 | 0.29±0.04 |

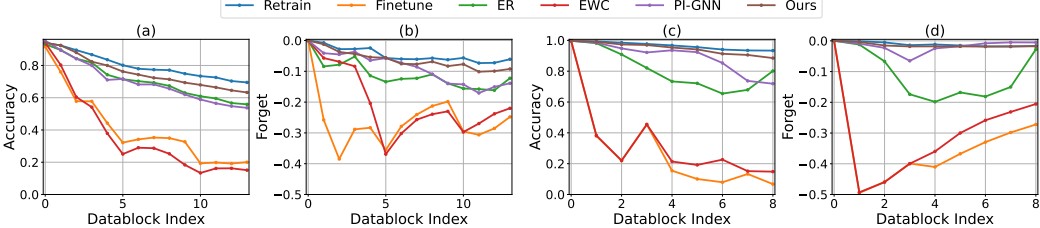

Figure 4: Performance progression over data blocks of our models and baselines. (a) CoraFull Accuracy. (b) CoraFull Forget. (c) Reddit Accuracy. (d) Reddit Forget.

we see that the dense version usually outperforms the sparse version, but the sparse version still achieves highly competitive results compared to the dense version and other baselines.

We also observe a similar pattern in the instance incremental setting in Table 2, where our model performs better than baselines on most datasets. Meanwhile, we acknowledge that the performance improvement on instance incremental datasets is not as significant as in the class incremental setting. Note that the lower bound online model also achieves comparable performance, indicating that forgetting was not a severe issue in these datasets. Hence, our method's advantage is not apparent.

Table 3: Training time comparison (Seconds/Epoch).

| | CoraFull | Arxiv | Reddit |
|---|---|---|---|
| Finetune | 2.30 | 3.38 | 8.41 |
| ER-GNN | 2.36 | 3.94 | 9.57 |
| Retrain | 6.03 | 16.37 | 28.68 |
| DyMoE(k=3) | 2.47 | 4.29 | 10.35 |
| DyMoe | 2.69 | 7.83 | 14.91 |

In Table 3, we show the training time of the baselines and our model. Compared to the retrain baseline (performance upper-bound), our model costs significantly less running time, only 34.4% on average, and we achieved competitive results as shown in Table 1 and Table 2. While the training time of our model is slightly higher than some baselines, our model obtains a remarkably better performance, showing that DyMoE is an economic trade-off between efficiency and effectiveness. We additionally provide inference time comparison in Appendix B.

Furthermore, we plot the AA and AF with respect to the data block sequences in Figure 4. Regularization-based methods struggle to keep learned information in the class incremental learning setting as its AF quickly increases. Our proposed DyMoE usually archives the closest performance with the upper-bound method (retrain baseline), while other baselines either fail to learn new information or forget old knowledge quickly.

## 5.2 INVESTIGATION OF DYMOE

To answer **Q2**, we compare our method with three other baselines, ER-GNN, PI-GNN, and C-GNN, that use memory nodes to help retain old knowledge. An ideal incremental learning method should only use a small memory size to obtain desirable performance. We plot the results with different memory sizes in Figure 5. From the results, we can see that our approach achieves better performance

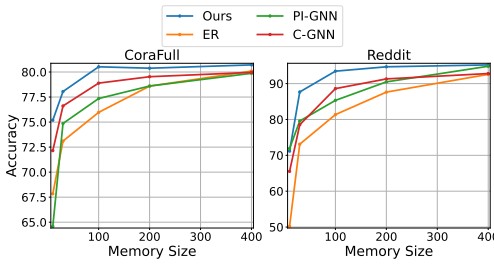

Figure 5: Model accuracy versus memory size for memory-based models.

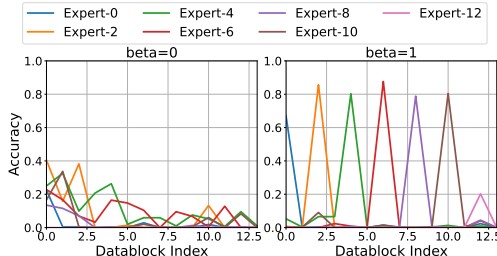

Figure 6: Individual experts' performances on dedicated data blocks.

Table 4: Ablation study of class incremental and instance incremental datasets.

|        | Reddit | | Cora-Full | | Paper100M | | Arxiv-CIL | |
|--------|--------|----|-----------|----|-----------|----|-----------|----|
|        | AA | AF | AA | AF | AA | AF | AA | AF |
| Full   | **93.28**$_{\pm 0.19}$ | **-2.98**$_{\pm 0.36}$ | 80.97$_{\pm 0.58}$ | -5.22$_{\pm 0.49}$ | **83.97**$_{\pm 1.16}$ | **-2.37**$_{\pm 0.83}$ | **68.06**$_{\pm 1.54}$ | -10.68$_{\pm 1.97}$ |
| Sparse | 91.57$_{\pm 0.58}$ | -3.46$_{\pm 0.36}$ | **81.33**$_{\pm 0.85}$ | -5.69$_{\pm 1.12}$ | 82.93$_{\pm 1.53}$ | -3.31$_{\pm 1.58}$ | 67.25$_{\pm 0.97}$ | **-9.54**$_{\pm 0.69}$ |
| w/o DB | 89.57$_{\pm 1.28}$ | -4.01$_{\pm 0.42}$ | 80.16$_{\pm 0.65}$ | **-4.52**$_{\pm 1.37}$ | 83.09$_{\pm 1.47}$ | -2.84$_{\pm 1.39}$ | 64.50$_{\pm 0.74}$ | -10.51$_{\pm 1.08}$ |
| w/o BL | 90.48$_{\pm 1.46}$ | -3.52$_{\pm 0.71}$ | 76.33$_{\pm 1.44}$ | -6.08$_{\pm 1.16}$ | 81.25$_{\pm 1.69}$ | -2.54$_{\pm 0.92}$ | 63.10$_{\pm 0.78}$ | -12.91$_{\pm 1.58}$ |
| w/o Dy | 92.09$_{\pm 1.08}$ | -2.99$_{\pm 0.74}$ | 78.67$_{\pm 0.71}$ | -5.41$_{\pm 1.02}$ | 81.50$_{\pm 1.09}$ | -3.73$_{\pm 1.32}$ | 64.30$_{\pm 0.85}$ | -14.52$_{\pm 1.77}$ |

with the same size of memory, especially when we only have 10 memory data points budget per data block. Moreover, we notice that DyMoE with 30 samples achieved comparable performance with ER-GNN and PI-GNN with 200 samples, which is roughly a 75% reduction in memory requirements. Note that when the memory size approaches infinity, all methods become retrained, and hence we are seeing a converging pattern for the baselines.

To answer **Q3**, we conduct an ablation study comparing the entire model, sparse model, sparse model initially containing the same number of experts as the data blocks (w/o Dy), sparse model without block-guided loss(w/o BL), and finally the sparse model without data balance training (w/o DB). The results are in Table 4. We see performance drop whenever a component is missing from the model, validating the importance of each component. In particular, we observe, in w/o Dynamic, that even though the model keeps a large parameter size since the first data block arrives, the performance still drops a lot. This shows that training experts in a sequence is the key to a successful model, with the ability to learn appropriate knowledge for each data block. Moreover, data balance training also helps overall performance, as it is crucial to reflect the actual distribution of the data blocks.

**Q4** validates whether our model and training procedure results in specialized experts as designed. We evaluate the performance of each expert on individual data blocks and plot the average accuracy of each data block. In Figure 6, we compare individual experts' performances varying the $\beta$. We can easily observe that when the beta is zero (block-guided loss is disabled), the experts are not specialized. The prediction relies on multiple experts, not necessarily the one corresponding to the target data. On the other hand, when trained with block-guided loss, the experts are specialized, and they achieve high prediction accuracy for their corresponding data blocks, validating our hypothesis that block-guided loss encourages higher levels of specialization.

# 6 CONCLUSION, LIMITATIONS, AND FUTURE WORK

In this paper, we identified the drawbacks of existing graph incremental learning models and proposed the DyMoE module with a sparse version to model different interaction types between data blocks effectively and efficiently. However, we also acknowledge that our model may have trouble locating the correct experts when there are too many data blocks, resulting in compromised performance. While this can be solved by periodic retraining, we plan to extend our work to handle extremely long data sequences (over 1000 data blocks) in future work.

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

APPENDIX

# A    PROOF OF THEOREM 1

We restate Theorem 1 for completeness,

**Theorem 1.** *For an arbitrary continual learning problem, suppose a PI model obtains a cross-entropy loss $\mathcal{L}_{PI}$, there exists a parametrization of DyMoE that achieves cross-entropy loss $\mathcal{L}_{Dy} = \mathcal{L}_{PI}$. When the data sequence follow a mixture of Gaussian distribution, we have $\mathcal{L}_{Dy} \leq \mathcal{L}_{PI}$.*

It is easy to see that DyMoE is at least as powerful as PI since we can parameterize all gating vectors with the same value; hence, the weights of all experts are the same, which makes the final output essentially a summation of each expert's output. In this case, DyMoE degenerates to PI. We then prove that under the Gaussian Mixture assumption of the data blocks, DyMoE achieves lower loss and, hence, is strictly more powerful than PI.

*Proof.* Consider the case with two data blocks generated from Gaussian Distributions, $X_1 = \mathcal{N}(\boldsymbol{\mu}_1, \sigma^2 I)$, and $X_2 = \mathcal{N}(\boldsymbol{\mu}_2, \sigma^2 I)$. For simplicity, we assume the same variance across coordinates and the probability of data from each distribution is the same. Let the distance between two distributions be $B$. The labels of data are generated depending on their distance from the mean of their coorresponding distribution, specifically, if $\boldsymbol{x} \sim X_1$,

$$y = \begin{cases} 0, & \text{if } ||\boldsymbol{x} - \boldsymbol{\mu}_1|| \leq d \\ 1, & \text{otherwise} \end{cases} \tag{14}$$

and if $x \sim X_1$,

$$y = \begin{cases} 2, & \text{if } ||\boldsymbol{x} - \boldsymbol{\mu}_2|| \leq d \\ 3, & \text{otherwise} \end{cases} \tag{15}$$

where $2d \leq B$ is a threshold distance to determine the data labels. This is a practical assumption for a mixture of two Gaussian distributions.

We then consider the procedure of Parameter Isolation (PI) and our proposed method. PI first trains a model $f_1(x)$ on $X_1$ and then trains a model $f_2(x)$ on $X_2$, both $f_1$ and $f_2$ are in $R^4$ for the four target classes. Hence, when making predictions, we have the logits to be:

$$y = \text{softmax}(f_1(x) + f_2(x)) \tag{16}$$

Since our approach can initialize a network with the same architecture, we can have the same network and parameters as the ones in PI, and the predictions from our model are:

$$y = \text{softmax}(\alpha_1 f_1(\boldsymbol{x}) + \alpha_2 f_2(\boldsymbol{x})), \quad \alpha_i = \frac{exp(-\frac{||\boldsymbol{x} - \boldsymbol{g}_i||^2}{2\sigma^2})}{exp(-\frac{||\boldsymbol{x} - \boldsymbol{g}_1||^2}{2\sigma^2}) + exp(-\frac{||\boldsymbol{x} - \boldsymbol{g}_2||^2}{2\sigma^2})} \tag{17}$$

Here, we use the negative of the distance normalized by the variance as the similarity measure between the input and the gating vectors. This is a valid and tractable choice as variance can be estimated by batch normalization. Note that in the first data block, we directly set the gating vector to the empirical mean of the input, $\boldsymbol{g}_1 = \bar{\boldsymbol{x}}_1 \approx \boldsymbol{\mu}_1$. In the second data block, the block-guided loss solves the problem

$$\min_{\boldsymbol{g}_2} \frac{1}{N} \sum_{i=1}^{N} s(\boldsymbol{x}_i, \boldsymbol{g}_2) \tag{18}$$

which is minimized by $\boldsymbol{g}_2 = \boldsymbol{\mu}_2$. We can then rewrite the prediction of the model:

$$y = \text{softmax}(\alpha_1 f_1(\boldsymbol{x}) + \alpha_2 f_2(\boldsymbol{x})), \quad \alpha_i = \frac{exp(-\frac{||\boldsymbol{x} - \boldsymbol{\mu}_i||^2}{2\sigma^2})}{exp(-\frac{||\boldsymbol{x} - \boldsymbol{\mu}_1||^2}{2\sigma^2}) + exp(-\frac{||\boldsymbol{x} - \boldsymbol{\mu}_2||^2}{2\sigma^2})} \tag{19}$$

To show that our model achieves lower loss on this task, we only need to consider the expected loss on the $D_1$ as the two distributions are symmetric. We divide the problem into two cases $||\boldsymbol{x} - \boldsymbol{\mu}_1|| \leq d$, when the input is close to the distribution mean, and $||\boldsymbol{x} - \boldsymbol{\mu}_1|| > d$ when the input is farther away.

For $||\boldsymbol{x} - \boldsymbol{\mu}_1|| \leq d$, the correct label is 0. We consider the cross-entropy loss of PI,

$$\mathcal{L}_{PI} = -log(\frac{f_1(\boldsymbol{x})_0}{f_1(\boldsymbol{x})_0 + f_1(\boldsymbol{x})_1 + f_2(\boldsymbol{x})_2 + f_2(\boldsymbol{x})_3})$$

$$= -log(\frac{\alpha_1 f_1(\boldsymbol{x})_0}{\alpha_1 f_1(\boldsymbol{x})_0 + \alpha_1 f_1(\boldsymbol{x})_1 + \alpha_1 f_2(\boldsymbol{x})_2 + \alpha_1 f_2(\boldsymbol{x})_3})$$
(20)

and the cross-entropy loss of our method

$$\mathcal{L}_{Dy} = -log(\frac{\alpha_1 f_1(\boldsymbol{x})_0}{\alpha_1 f_1(\boldsymbol{x})_0 + \alpha_1 f_1(\boldsymbol{x})_1 + \alpha_2 f_2(\boldsymbol{x})_2 + \alpha_2 f_2(\boldsymbol{x})_3})$$
(21)

since $||\boldsymbol{x} - \boldsymbol{\mu}_1|| \leq d \leq B - d \leq ||\boldsymbol{x} - \boldsymbol{\mu}_2||$, meaning that $\alpha_1 \geq \alpha_2$, and we have $\mathcal{L}_{Dy} \leq \mathcal{L}_{PI}$. Since cross-entropy is monotonic, we can obtain the minimum of $\mathcal{L}_{PI} - \mathcal{L}_{Dy}$ at $||\boldsymbol{x} - \boldsymbol{\mu}_1|| = d$ and $||\boldsymbol{x} - \boldsymbol{\mu}_2|| = B - d$. Let the minimum be $\Delta\mathcal{L}_{close}$. Note $\Delta\mathcal{L}_{close}$ increases as $d$ increases and $\sigma$ decreases.

For $||\boldsymbol{x} - \boldsymbol{\mu}_1|| > d$, the correct label is 1. Let $M$ be the maximum absolute value that the neural network $f_2$ output for a logit, that is $|f_2(x)_y| \leq M$. Then, the maximum possible loss is $-f_2(x)_1 + log(C \cdot exp(M)) = log(C) + 2M$, where $C = 4$, the number of classes. Since logits of PI and our method is bounded by the same $M$, we have the maximum possible loss difference to be

$$|\Delta\mathcal{L}_{far}| = 4M + 2logC$$
(22)

We now developed a lower bound for the loss difference when $\boldsymbol{x}$ is close to $\boldsymbol{\mu}_1$ and an upper bound for the loss difference when x is far from $\boldsymbol{\mu}_2$, we then compute the probability of each case using Gaussian tail bound.

$$P[\boldsymbol{x} - \boldsymbol{\mu}_1 > d] \leq exp(-\frac{d^2}{2\sigma^2}), P[\boldsymbol{x} - \boldsymbol{\mu_1} \leq d] \geq 1 - exp(-\frac{d^2}{2\sigma^2})$$
(23)

Then the upper bound of the difference in expected loss when $x$ is far is:

$$|\Delta E_{far}| \leq (4M + 2logC) \cdot exp(-\frac{d^2}{2\sigma^2})$$
(24)

The lower bound of the expected loss when $x$ is close is:

$$\Delta E_{close} \geq \Delta\mathcal{L}_{close} \cdot (1 - exp(-\frac{d^2}{2\sigma^2}))$$
(25)

Taking the ratio:

$$\frac{|\Delta E_{far}|}{\Delta E_{close}} \leq \frac{(4M + 2logC) \cdot exp(-\frac{d^2}{2\sigma^2})}{\Delta\mathcal{L}_{close} \cdot (1 - exp(-\frac{d^2}{2\sigma^2}))}$$
(26)

As $\frac{d}{\sigma}$ increases, the ratio approaches zero, hence we have the overall expected loss difference,

$$\Delta E = \Delta E_{close} + |\Delta E_{far}| \geq \Delta E_{close} - |\Delta E_{far}| \geq 0$$
(27)

making the overall loss difference positive, and our approach leads to lower loss in this case. $\qquad\square$

## B    INFERENCE TIME EXPERIMENT

We additionally provide an inference time comparison between our method and a strong baseline ER-GNN. We report the inference time on the last inference epoch of the training data sequences, and the numbers of samples for all methods are the same. For a fair comparison, we report DyMoE, DyMoE with $k = 3$, ER-GNN, and ER-GNN ($\times 3$), where the size of the ER-GNN is enlarged roughly three times to have a similar number of active parameters as the sparse DyMoE. The results are in Table 5. We can see that while the inference time of DyMoE is slightly worse than ER-GNN, we achieved much better performance, showing the necessity of the specialized experts. Comparing the enlarged version of ER-GNN, we see that simply increasing the model size does not benefit the performance despite higher computation time, which further illustrates the advantage of the proposed framework to properly use the extra trainable parameters.

Table 5: Inference time comparison (Seconds/Epoch for time).

|  | CoraFull | | Arxiv-CIL | | Reddit | |
|---|---|---|---|---|---|---|
|  | Time | AA | Time | AA | Time | AA |
| ER-GNN | 2.89 | 71.08 | 11.07 | 57.09 | 18.41 | 81.35 |
| ER-GNN ($\times$3) | 3.48 | 70.54 | 14.61 | 59.44 | 19.89 | 80.69 |
| DyMoE(k=3) | 3.50 | 81.33 | 13.65 | 67.25 | 20.55 | 91.57 |
| DyMoe | 4.13 | 80.97 | 18.77 | 68.06 | 24.39 | 93.28 |

Table 6: Hyperparameters for class incremental learning.

|  | Arxiv-CIL | DBLP-CIL | CoraFull | Reddit |
|---|---|---|---|---|
| Learning Rate | | 0.0001 | | |
| Weight Decay | | {0.01, **0.001**, 0.0001} | | |
| Embedding Dimension | 512 | 128 | 128 | 128 |
| # Epochs | | 40 | | |
| # Balancing Epochs | | 10 | | |
| $\beta$ | {0.01, 0.1, **1**, 2} | {0.01, **0.1**, 1, 2} | {0.01, 0.1, **1**, 2} | {0.01, 0.1, **1**, 2} |
| batch size | | 128 | | |

## C EXPERIMENT DETAILS

### C.1 IMPLEMENTATION DETAILS

The model is implemented in PyTorch and DGL, and all experiments are conducted on 1 Nvidia Tesla T4 GPU. We repeat the experiment 5 times using different random seeds and report the mean and standard deviation. We uniformly use a fan-out of 10 to extract subgraphs from each target node. The hyperparameters used during training are shown in Table 6 and Table 7, where the curly bracket represents the hyperparameters for searching, and the hyperparameters selected are marked in bold. Memory size is the per data block memory size, and it is a special hyperparameter in the continual learning setting because as it increases, all methods converge to the retrain method, which is usually the upper bound of all continual learning methods. Hence, we set a fixed memory size at 100 for our model, and for a fair comparison, if the baseline model requires a memory set, we use the same number.

### C.2 DATASET DETAILS

The dataset statistics are shown in Table 8. We collect data from academic graphs (Arxiv, DBLP, Paper100M, CoraFull), social networks (Reddit), and BlockChain networks (Elliptic) to show that our model handles a wide range of datasets. We describe the construction of each dataset as follows and includes the number of new nodes and edges in Figure 7 and 8.

**ArXiv**: Arxiv academic citation network from Open Graph Benchmark (OGB) (Hu et al., 2021) contains arxiv articles and the citation information between articles. For instance incremental learning setting, we use the first 25 timestamps in the original arxiv dataset as the first data block, as they contain significantly less data. We then split the rest of the data by year, and data in each forms a data block. For class incremental learning setting, we split the data into 8 blocks each contains 5 classes.

**DBLP**: DBLP is an academic network from the DBLP website containing computer science academic paper, with citation information (Tang et al., 2008). We follow Zhang et al. (2023a) to sample 20000 nodes with 9 classes and 75706 edges from DBLP full data, we split it into data blocks according to the timestamps. For the class incremental setting, we split the 9 classes into 5 data block each containing 2 classes, except for the last one with only 1 class.

**Paper100M**: Paper100M is a citation network extracted from Microsoft Academic Graph by OGB (Hu et al., 2021). We follow Zhang et al. (2023a) to sample 12 classes from the year 2009 to the year 2019 from Paper100M full data and we split it into tasks according to the timestamps.

Table 7: Hyperparameters for instance incremental learning.

|  | Arxiv-IIL | DBLP-IIL | Paper100M | Elliptic |
|---|---|---|---|---|
| Learning Rate | | 0.0001 | | |
| Weight Decay | | 0.001 | | |
| Embedding Dimension | 512 | 128 | 256 | 128 |
| # Epochs | | 40 | | |
| # Balancing Epochs | | 5 | | |
| $\beta$ | $\{0.01, 0.1, \mathbf{1}, 2\}$ | $\{\mathbf{0.01}, 0.1, 1, 2\}$ | $\{0.01, \mathbf{0.1}, 1, 2\}$ | $\{0.01, \mathbf{0.1}, 1, 2\}$ |
| batch size | | 128 | | |

Table 8: Dataset statistics.

|  | #. Nodes | #. Edges | #. Classes | #. Data blocks | #. Classes per block |
|---|---|---|---|---|---|
| CoraFull | 19793 | 126842 | 70 | 14 | 5 |
| Arxiv-CIL | 169343 | 2332486 | 40 | 8 | 5 |
| Reddit | 232965 | 114615892 | 41 | 9 | 5 |
| DBLP-small-CIL | 20000 | 302862 | 9 | 5 | 2 |
| Paper100M-small | 49459 | 217420 | 12 | 11 | NA |
| Arxiv-IIL | 169343 | 2332486 | 40 | 11 | NA |
| DBLP-small-IIL | 20000 | 302826 | 9 | 24 | NA |
| Elliptic | 203769 | 468710 | 2 | 49 | NA |

**CoraFull**: CoraFull is a co-citation academic network, where nodes are papers, and the two nodes are connected if they are co-cited by other papers (Bojchevski & Günnemann, 2018). We use the provided CoraFull data from DGL, and split its 70 classes into 14 5-classes data blocks for class incremental learning.

**Reddit**: The Reddit dataset contains Reddit posts as nodes, and two nodes are connected by edges if they are posted by the same user (Hamilton et al., 2017). We use the provided Reddit data from DGL, and split its 40 classes into 8 5-classes data blocks for class incremental learning.

**Elliptic**: The Elliptic dataset is a bitcoin transaction network, where each node represents a transaction, and each edge denotes money flow (Weber et al., 2019). Its nodes have timestamps evenly spaced with an interval about two weeks. We use the original timestamp from the dataset for instance-incremental learning.

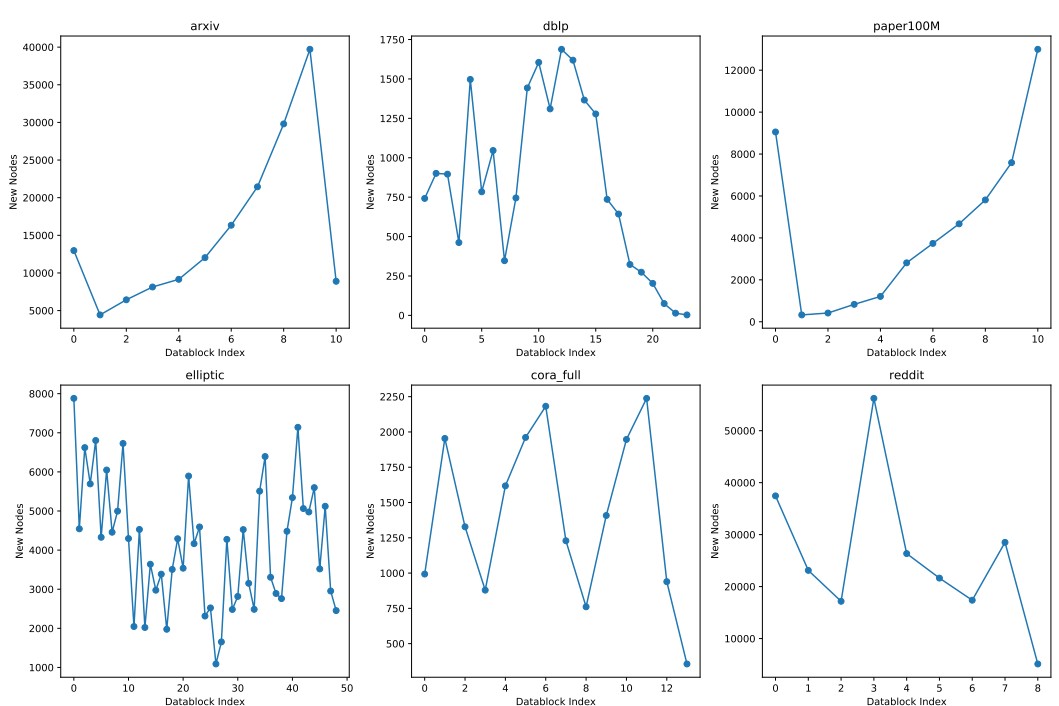

Figure 7: Number of new nodes per data block.

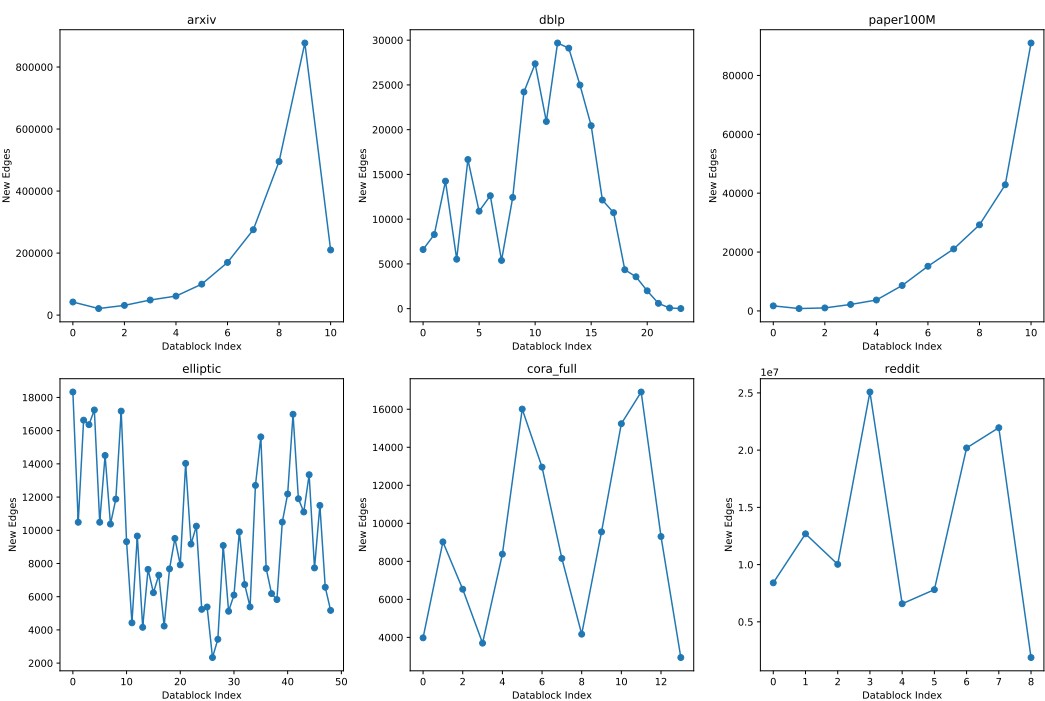

Figure 8: Number of new edges per data block.

