# OpenReview forum: "Dynamic Mixture-of-Experts for Incremental Graph Learning"
_ICLR.cc/2025/Conference — Submitted to ICLR 2025_

### Official Review · Reviewer_6WFG · 2024-10-28

**Soundness:** 3
**Presentation:** 3
**Contribution:** 1
**Rating:** 3
**Confidence:** 4

**Summary:**

The paper presents a Dynamic Mixture-of-Experts (DyMoE) framework for incremental graph learning, aiming to address catastrophic forgetting in graph neural networks (GNNs) when new data arrives sequentially. Unlike traditional approaches that treat prior knowledge uniformly, DyMoE dynamically adds specialized expert networks to handle each new data block, optimizing the reuse of relevant information. To reduce computational demands, the authors introduce a sparse DyMoE variant that selectively activates only the top-k experts for predictions. DyMoE achieves a notable improvement in accuracy with minimal computational overhead, making it an effective solution for class and instance incremental learning settings.

**Strengths:**

- the studied problem is important and interesting. It deserves more attention
- the proposed method and identified problem, despite not being very novel, is sound to some extend.

**Weaknesses:**

- the identified problem and the proposed method are not novel.

  a. the correlation induced by the dependency of graph data was previously introduced (e.g., see [1,2]), rendering continual learning on graph data is different from other i.i.d data

  b. applying the MoE model to continual learning seems to be a common combination (e.g., see [3,4]). I certainly agree with the fact that in different settings (especially under graph data), such a combination might require different considerations and tailored designs. However, in the current version of the paper, I fail to understand what are the unique problems for applying such a combination in the studied setting in the paper.

- the effectiveness of the method needs further support

  a. I agree with the intuition that a separate expert network can better store/memorize information for each task. However, it comes with two natural questions: 1) what happens to the parameter complexity (overall model size) when the data stream is large and 2) how effective is this approach for parameter/information sharing among tasks (otherwise, why not train a separate network for each task)

  b. It is not clear why the gating mechanism in MoE is effective in addressing catastrophic forgetting. In the graph data case, as shown in [1,2], the correlation/dependence among data can induce a distribution shift and can render the problem unlearnable. Why the (proposed) gating mechanism can address this? Furthermore, even in the case of i.i.d data, it is still not clear to me why the gating mechanism does not suffer from catastrophic forgetting (e.g., forget how to router the decision for the previous task).

- the theoretical result is not rigorous and clear, and it is not connected well to support the proposed method

  a. First, it is not clear what $\mathcal{L}(Dy)$ and $\mathcal{L}(PI)$ are exactly. Assuming the loss function is the one given by (10), the loss function would have a dependency on the $\beta$ which is not reflected in the theorem statement.

  b. There are many cases, where the theoretical result does not add any value. For example, in the case of separable data, it can be shown that with enough model parameters, there exists a model parameter that can achieve zero loss. In this case, the other side of the result is also true, i.e., $\mathcal{L}(PI) \leq \mathcal{L}(Dy)$. Furthermore, the result is about existence. It does not tell us much about how difficult to find such a parameter.

  c. The theorem statement tells us that it is possible for the proposed method to achieve a smaller loss value (with respect to the proposed objective) compared to the parameter isolation. It needs further argument why the proposed training objective is the best/very reasonable for measuring model performance. what happens if the objective is switched to the one used in $\mathcal{L}(PI)$ paper? How does this training loss connect with the generalization performance (what we actually care)?

- the experimental studies have some questions as well

  a. (minor) The selected baselines seem to be a bit outdated

  b. (major) The performance of the baseline method seems to be much lower than the one presented in the benchmark study[5]. What are the reasons behind this?

[1] "Towards robust graph incremental learning on evolving graphs." International Conference on Machine Learning. PMLR, 2023.

[2] "Continual Learning on Graphs: A Survey." arXiv preprint arXiv:2402.06330 (2024).

[3] "Theory on Mixture-of-Experts in Continual Learning." arXiv preprint arXiv:2406.16437 (2024)

[4] "Boosting continual learning of vision-language models via mixture-of-experts adapters." Proceedings of the IEEE/CVF Conference on Computer Vision and Pattern Recognition. 2024.

[5] "Cglb: Benchmark tasks for continual graph learning." Advances in Neural Information Processing Systems 35 (2022): 13006-13021.

**Questions:**

see weakness

---

> ### Author Response · Authors · 2024-11-20
> **Author response 1**
>
> Thank you for your very knowledgeable and thoughtful comments, these greatly help us in making our paper better, and we would like to discuss some of your concerns here.
>
> > The identified problem and the proposed method are not novel.
>
> The continual learning problem in the graph is unique compared to that in the CV/NLP domain. Specifically, the model not only needs to learn from new data blocks and keep the old knowledge but also needs to adapt the representation learned for old blocks according to the new graph contexts, as the new data block may change the topology of nodes from the old data blocks.
>
> There are two challenges in graph continual learning: 1) Reducing forgetting: The model needs to remember old data blocks, which is a common challenge for other continual learning domains, like CV and NLP.  2) Adapting old nodes to new graph context: unlike CV/NLP, the nodes in old data blocks change as new data arrives, a model should account for this.
>
> To solve the first challenge, we dynamically add MoE modules to learn individual data blocks to reduce forgetting. The trained modules are not modified afterward, preserving as much learned information as possible. Consider an experience-replay variant that finetune the model with the memory node set from previous data blocks, it can potentially modify all parameters of the model causing significant forgetting, whereas we only adjust the gating vectors, meaning that we can replicate the output when the gating values are correct. Lastly, to ensure the gating value correctness, we propose the block-guided loss, which is the directly supervised signal to produce correct gating values.
>
> The second challenge happens because a node from data block 1 can later be connected to nodes from future data blocks. Our design interleaves the MoE modules into GNN layers, so the models learn to calibrate the input to the next layer via message-passing, without changing the experts' parameters. Intuitively, because of the distribution shift, the previously trained experts no longer work. Training like ER-GNN would significantly change the expert’s performance while heavily overfitted towards the memory nodes, whereas we only update the gating vectors to account for the distribution shift in the representative memory nodes and all information previously about the entire data block is not lost.
>
> While MoE in CV/NLP are indeed used for continual learning, they do not tackle the second challenge. For example, to perform class-incremental learning, [1] needs to train an auto-selector which depends on a specific data block, and does not evolve over time. Consequently, if we adapt this method to the graph domain, an auto-selector trained for data block one will fail quickly when new data blocks arrive, because the auto-selector no longer works for data block one as the new connection changes the distribution of the nodes in data block one.
>
> > What happens to the parameter complexity (overall model size) when the data stream is large
>
> The overall complexity indeed increases linearly as the length of the data stream increases, which is why we implemented the sparse variant to ensure the activated parameters and inference time remains constant as the length increases.
>
> > How effective is this approach for parameter/information sharing among tasks (otherwise, why not train a separate network for each task)
>
> An important target in our paper is a class-incremental setting, where new data blocks introduce new classes and the tasks also require the model to distinguish these new classes from the old ones. A separate model cannot perform this task, and hence an integrated model is essential. Meanwhile, to study the effectiveness of information sharing we additionally conduct task-incremental learning.
>
> |             | Cora-full | Arxiv-IIL |
> |-------------|-----------|-----------|
> | Separate    | 84.61     | 63.19     |
> | ER-GNN      | 83.21     | 66.53     |
> | C-GNN       | 85.94     | 67.28     |
> | DyMoE (k=3) | 87.82     | 67.76     |
>
> We compare DyMoE to the separate baseline where we train separate models for each task. We can see that without support from a larger dataset (previous data blocks), the separate model struggles to achieve optimal performance.

---

> ### Author Response · Authors · 2024-11-20
> **Author response 2**
>
> > The effectiveness of the gating mechanism.
>
> Our framework is more than the MoE architecture, it contains a dynamically added MoE module with a **gating/router mechanism**, a collected memory dataset containing subsets of previously trained data, and block-guided loss to ensure old data in the memory set is still routed correctly to their corresponding experts. Specifically, we keep a small memory set of previous data blocks, when new data arrives, adding new knowledge and changing old patterns, the memory node-set is used as a proxy of the entire old data block. The detailed information of the old data block is stored within each frozen expert ensuring the knowledge is not lost, whereas we adapt the gating process through training with the memory node set to account for the overall distribution shift.
> In terms of unlearnability, could you be a bit more specific about the conclusion? In [2], the authors developed a catastrophic forgetting upper bound consisting of the training loss, an intra-block structural shift loss, and an inter-block structural shift loss for a **fixed** GNN. This bound only correlates to the performance of a fixed GNN under structural shift and does not suggest unlearnable results. To the best of my understanding, their proposed method shows that by properly training the model, they can minimize such a loss to improve continual learning performance.  Intuitively, consider two data blocks, whose structures follow completely different distributions and are isolated graph components, which results in high inter-block structural shift. In DyMoE, the first expert is trained on one block, and its gating vector is set as the mean of the inputs. The input to the MoE module will be very different for data in the second block because of the structural shift, and the second expert learns to capture that with a gating vector very different from the first gating vectors. Thus during inference, the input that is closer to the first gating vector will have high gating values, and correctly route the input.
>
> > First, it is not clear what $\mathcal{L}(Dy)$ and $\mathcal{L}(PI)$ are exactly. Assuming the loss function is the one given by (10), the loss function would have a dependency on the $\beta$ which is not reflected in the theorem statement.
>
> For the theoretical analysis, we follow a clean setting, where the loss is just the cross entropy, which directly reflects the prediction accuracy. The block-guided loss $\mathcal{L}_{BL}$ is omitted, because it is an auxiliary loss for regularization not directly related to the prediction accuracy, and it would be pointless to include this loss when we compare PI with our method.
>
> > There are many cases, where the theoretical result does not add any value. For example...
>
> The proof has two parts, and the first part is easy to see, DyMoE is always at least as good as PI. To find such a parametrization, the model can simply set all gating vectors to the same one. The second part is to show that under certain conditions (Gaussian), DyMoE can achieve better loss, which shows that DyMoE is a stronger model. Note that in training, the block-guided loss will enforce inputs to be close to their gating vectors, which encourages the gating vectors to stay at the mean of the inputs.
>
> In terms of separable data, consider both PI and DyMoE, models can indeed achieve zero loss in a fully supervised setting, but in the continual learning setting, when the model is presented with only the second data block, the incremental model/expert will optimize only for the second data-block, leading to increased loss for data from the first data block, because the incremental model will only output large logits for classes in the second data block. The proof in the paper shows that for any loss achieved by the PI model, DyMoE can achieve a lower one. To achieve such a lower loss, we only need the gating vector to be close to the distribution mean of their corresponding data block, which is achieved by the block-guided loss. We will add more clarity to both the theorem statement and intuition.

---

> ### Author Response · Authors · 2024-11-20
> **Author response 3**
>
> > The theorem statement tells us that it is possible for the proposed method to achieve a smaller...
>
> As we discussed in your previous concern, the loss is just the classification loss (cross-entropy) between the predicted value and the true label, as can be seen in equation (20), we will state this more clearly in the theorem. So here the loss is independent of the model and directly reflects the accuracy (if a model has low cross-entropy loss, it assigns high probability to correct classes, and hence higher accuracy). Optimizing the cross-entropy loss will optimize the performance. The theorem proves that, in arbitrary circumstances, DyMoE can achieve the same classification loss, which is a direct indicator of the classification accuracy, as PI. And, in the case of a Mixture of Gaussian, DyMoE can achieve lower classification loss, leading to higher accuracy.
>
> > Outdated baselines.
>
> We compare DyMoE with more recent baselines, including RCL-CN[3] and SSRM[2] in general response. DyMoE is still strong compared to these baselines.
>
> > Inconsistent performance.
>
> The lower performance is because we follow a different split than the CGLB[4]. We follow the split in the PI-GNN[5] paper.
>
> [1] "Boosting continual learning of vision-language models via mixture-of-experts adapters." Proceedings of the IEEE/CVF Conference on Computer Vision and Pattern Recognition. 2024.
>
> [2] "Towards robust graph incremental learning on evolving graphs." International Conference on Machine Learning. PMLR, 2023.
>
> [3] ''Reinforced continual learning for graphs,'' in Proceedings of the 31st ACM International Conference on Information & Knowledge Management, 2022, pp. 1666–1674.
>
> [4] "Cglb: Benchmark tasks for continual graph learning." Advances in Neural Information Processing Systems 35 (2022): 13006-13021.
>
> [5] "Continual learning on dynamic graphs via parameter isolation." Proceedings of the 46th International ACM SIGIR Conference on Research and Development in Information Retrieval. 2023.

---

> > ### Comment · Reviewer_6WFG · 2024-11-21
> >
> > Thank you for your diligent responses. Unfortunately, I still found most of my main concerns unresolved. Please allow me to summarize/rephrase them as follows.
> >
> > Main concerns:
> >
> > 1. one of the claimed contributions of the paper "identified the issue of existing continual learning methods that ignore the correlation between different data blocks". Please explain what the new insights presented in the paper and how they are different from the previous works on graph continual learning [1].
> >
> > 2. the integration between studied problem, methods and theoretical results
> >
> > a. I completely agree that modifications have been added to the MoE model. I do not see how the proposed design is tailored or motivated by the unique challenges faced by GCL. Can you explain the connection between the proposed design and the dependency/correlation of data in GCL?
> >
> > b. For the theoretical comparison between MoE and PI. Is cross-entropy alone a typical learning objective/setting for the PI frameworks?
> >
> > c. It is known that MoE in general are better at learning the mixture of distribution. So what is the novel insight brought by the theoretical analysis that is tailored to the proposed setting?
> >
> > Additional comment:
> >
> > 1. I do not think the incremental class setting is a valid/complete argument, as one can train a binary classification model for each class.
> >
> > 2. the argument for the separable data is also not valid/complete, as there can be cases where the first data block and second block live in orthogonal subspace and the learning process does not interfere.
> >
> > 3. the unlearnability results mentioned are a straightforward corollary from the result of [1] and the classic result in transfer learning [2].
> >
> > [1] "Towards robust graph incremental learning on evolving graphs." International Conference on Machine Learning. PMLR, 2023.
> >
> > [2] "A survey on domain adaptation theory: learning bounds and theoretical guarantees." arXiv preprint arXiv:2004.11829 (2020).

---

> > > ### Author Response · Authors · 2024-11-21
> > >
> > > Thank you very much for the prompt reply. To further clarify your main concerns,
> > >
> > > > What's the distinction between DyMoE and SSRM[1].
> > >
> > > SSRM [1] is indeed an exciting and solid work, and it addresses a closely related problem. The key distinction between SSRM and our work is how the structural shift is captured. In SSRM, it is captured implicitly through a regularization loss that quantifies the distribution difference between the original and updated neighborhood. In DyMoE, the structural shift is explicitly captured by interleaving MoE into the GNNs. Consider the case when a node in previous data blocks is connected to new nodes in new data blocks as shown in Figure 3 of the paper, the input to the k-th DyMoE GNN layer is the output of the (k-1)-th layer. If we use the old experts in (k-1)-th layer to process the new nodes and feed the output to k-th layer, the model will fail because the old experts were never trained on the new data, and consequently forget. Instead, we train a new expert in (k-1)-th layer, whose representation can help predict old nodes under connections to new data, mitigating its negative impact.
> > >
> > > > Why the design is necessary for GCL problems.
> > >
> > > Unlike in [2] (vision domain), where one data point belongs to one data block, in GCL, one data point can contain data from multiple data blocks because we consider a node and all of its neighbors. Consequently, in the graph domain, multiple/potentially unrelated experts need to be activated to learn one node. The MoE and interleaving design here explicitly captured this, all neighbor nodes for a target node are processed by their own corresponding experts, the experts are aligned to the representation that helps the prediction of the target node.
> > >
> > > >  For the theoretical comparison between MoE and PI. Is cross-entropy alone a typical learning objective/setting for the PI frameworks?
> > >
> > > PI does use other auxiliary loss during training, but the only meaningful loss during evaluation is the cross-entropy loss, and we are comparing that between PI and DyMoE.
> > >
> > > >  ... So what is the novel insight brought by the theoretical analysis that is tailored to the proposed setting?
> > >
> > > The proof here is tailored to the continual learning context, which shows concretely that DyMoE, an MoE-based continual learning method, could achieve lower loss than PI. Here I try to use a more intuitive way to explain its implication. For two data blocks, PI is first trained to achieve zero loss on the first data block and then trained to achieve zero loss on the second data block without information from the first data block. This process will introduce loss to the first data block, and we prove that DyMoE can mitigate this loss.
> > >
> > > We hope these responses can lift your concerns. We still greatly appreciate your time evaluating our paper and your very knowledgeable comments to help improve our work.
> > >
> > > [1] "Towards robust graph incremental learning on evolving graphs." International Conference on Machine Learning. PMLR, 2023.
> > >
> > > [2] "Boosting continual learning of vision-language models via mixture-of-experts adapters." Proceedings of the IEEE/CVF Conference on Computer Vision and Pattern Recognition. 2024.

---

> > > > ### Comment · Reviewer_6WFG · 2024-11-25
> > > >
> > > > I thank the author for the diligent response. However, I do not think I have got direct/sufficient answers to my previous concerns/questions. As such, I will keep the original score.

---

### Official Review · Reviewer_Ejrw · 2024-10-29

**Soundness:** 2
**Presentation:** 2
**Contribution:** 2
**Rating:** 5
**Confidence:** 4

**Summary:**

The paper proposes a Dynamic Mixture-of-Experts (DyMoE) framework to address the challenge of catastrophic forgetting in dynamic graphs for incremental graph learning. The DyMoE framework introduces a novel approach by dynamically adding specialized expert networks for each incoming data block. These expert networks are used selectively through a gating mechanism, which determines the relevance of each expert based on the input data.

**Strengths:**

1. The paper identified the issue of existing continual learning methods that ignore the correlation between different data blocks.
2. The paper tackled a significant dynamic graph problem in real-world scenarios, where data arrives incrementally, offering a scalable solution without the need for full dataset retraining.
3. The paper developed a DyMoE module with specialized experts for each data block and introduced a data block-guided loss to reduce negative interference among the experts.

**Weaknesses:**

1. The use of symbols is inconsistent, and the explanations lack clarity. Specifically:

   (a) What do m and n represent in Eq. (3)? Are they referring to the feature dimension or the number of nodes?

   (b) In Eq. (13), what is the specific meaning of y? Does it convey the same meaning as h?

   (c) In Figure 1, triangles are used on the left and circles on the bottom right to represent blocks.

The authors can include a notation table or provide more explicit definitions for each symbol when first introduced.

2. The baseline methods used in the experiment mainly come from older works, with only one baseline in recent 3 years. State-of-the-art approaches, such as MSCGL [1], RLC-CN [2], SEM [3], and UGCL [4], would have provided a more comprehensive evaluation. Can you justify the choice of baselines if more recent methods were intentionally excluded?

   [1] J. Cai, X. Wang, C. Guan, Y. Tang, J. Xu, B. Zhong, and W. Zhu, ''Multimodal continual graph learning with neural architecture
       search,'' in Proceedings of the ACM Web Conference, 2022, pp.1292–1300.

   [2] A. Rakaraddi, L. Siew Kei, M. Pratama, and M. De Carvalho, ''Reinforced continual learning for graphs,'' in Proceedings of the
       31st ACM International Conference on Information & Knowledge Management, 2022, pp. 1666–1674.

   [3] Zhang, Xikun, Dongjin Song, and Dacheng Tao. ''Ricci curvature-based graph sparsification for continual graph representation
   learning,'' IEEE Transactions on Neural Networks and Learning Systems (2023).

   [4] T. D. Hoang, D. V. Tung, D.-H. Nguyen, B.-S. Nguyen, H. H.Nguyen, and H. Le, ''Universal graph continual learning,'' Transactions
   on Machine Learning Research, 2023.

3. In line 255, ''Figure 6 shows that direct training will not result in specialized experts'' seems confusing, can you explain it more clearly?

4. In Section 3.3, it’s unclear how to tweak the gating values during training randomly so all experts have similar selection chances. What is the magnitude of the tweaks, or can you give a mathematical formulation of how the randomization is applied?

**Questions:**

See weaknesses.

---

> ### Author Response · Authors · 2024-11-20
>
> Your comments are invaluable to us, and we will use them as important source to improve our work. Here, we would like to discuss some of your concerns.
>
> > The use of symbols is inconsistent
>
> The $n$ and $m$ are input and output dimensions, we’ve clarified this in the updated pdf.
>
> You are correct that $y$ is $h$, thanks for pointing this out, we’ve fixed it in the pdf.
>
> Here, the triangle and the vertical bar represent data blocks, not nodes. We use color and shapes to distinguish different data blocks.
>
> > More baselines.
>
> These are definitely related works and we will include them as important reference points in our revision. Meanwhile, we compared DyMoE to RCL-CN and SSRM which have open-sourced access in the general response.
>
> > In line 255, ''Figure 6 shows that direct training will not result in specialized experts'' seems confusing, can you explain it more clearly?
>
> This section motivates the design of block-guided loss. In Figure 6, we show two results of two models, the left one shows the results of a DyMoE model trained without block-guided loss, and the right one is trained with block-guided loss. In this experiment, after training, we isolate the experts by only activating one expert during inference to obtain their individual performance on data blocks. Every line in Figure 6 represents an expert’s isolated performance. We can see that if we train the MoE without block-guided loss, the experts are not specialized to have a good performance on their corresponding data blocks.
>
> > Randomization process.
>
> Here we follow the Sparse MoE [1]. Specifically, during training, the gating value is $g = \textbf{x}\cdot v + StandardNormal()\ast softplus(\textbf{x}\cdot v_n)$, where $v$ is the gating vector and $v_n$ is a trainable noise vector determining the level of noise.
>
> [1] Shazeer, Noam, et al. "Outrageously large neural networks: The sparsely-gated mixture-of-experts layer." arXiv preprint arXiv:1701.06538 (2017).

---

> ### Comment · Reviewer_Ejrw · 2024-11-21
>
> Thank you for your reply. For weaknesses 3 and 4, I still have two questions.
>
> Regarding Figure 6, I noticed that in the experimental section, the authors use it to "evaluate whether their model and training procedure result in specialized experts as designed." However, the author mentioned Figure 6 "motivates the design of block-guided loss". This creates a circular reasoning issue, as the **validation** of the method is being used as its **motivation**.
>
> Regarding the randomization process, Sparse MoE states that "the noise term helps with load balancing," while the authors claim that "all experts have similar selection opportunities." However, in the subsequent experiments, k=3. I am curious how this random noise ensures the inclusion of the last expert, as it seems that this randomness does not guarantee the last expert will be part of the top-k selection.

---

> > ### Author Response · Authors · 2024-11-21
> >
> > Thank you very much for your prompt reply! To answer your questions,
> >
> > > This creates a circular reasoning issue, as the validation of the method is being used as its motivation.
> >
> > We apologize if our description is not clear in the original response. A better way to explain the motivation is that: first, we want experts to specialize in corresponding data blocks so that as long as we can have correct gating vectors, we will have the correct results (low forget), and hence we have the dynamic MoE method to add and train a new expert, while freezing the rest. However, we found that in our preliminary study, we found this approach does not results in specialized experts as expected, so we further propose block-guided loss which eventually yield specialized experts.
> >
> > > How is the last expert selected?
> >
> > The last expert is just like all previous experts, we will apply the same random tweak to its gating values. Indeed, this mechanism does not guarantee the selection of the last expert. However, always selecting the last expert is not the desired behavior. For example, if a sample (one from the memory set) needs previous experts, selecting the last expert might not be the most beneficial choice for the performance. The randomization is not designed to always select the last expert, rather it ensures that the last (and new) expert is selected with a decent likelihood, so it's properly trained.
> >
> > Hope this response addresses your concerns, and we are happy to explain further.

---

> > > ### Comment · Reviewer_Ejrw · 2024-11-25
> > >
> > > Thank you to the authors for their detailed response. However, based on the responses provided by the authors, I remain unconvinced about two critical points:
> > >
> > > 1. Validation & Motivation: The explanation regarding the motivation for block-guided loss still lacks clarity and consistency. While the authors attempted to clarify their reasoning, it remains unclear how Figure 6 can serve both as a motivation and a validation. This circular reasoning issue has not been fully addressed, and I strongly encourage the authors to explicitly separate these concepts in their manuscript to avoid further confusion.
> > >
> > > 2. Randomization Mechanism: The authors' response regarding the random selection of experts provides some insights but lacks empirical evidence to substantiate their claims. Specifically, while they argue that all experts, including the last one, are selected with sufficient likelihood, they do not provide statistical analysis or experimental results to confirm this. Without concrete evidence, it is difficult to assess whether the mechanism achieves the desired balance or unintentionally neglects certain experts.
> > >
> > > Given these unresolved issues, I maintain my original score.

---

### Official Review · Reviewer_VebG · 2024-11-03

**Soundness:** 2
**Presentation:** 3
**Contribution:** 3
**Rating:** 5
**Confidence:** 4

**Summary:**

This paper proposes a dynamic mixture-of-expert (DyMoE) model for graph incremental learning. Specifically, DyMoE uses separate expert networks to model different data blocks. When a new data block arrives, it learns a new expert without modifying previously learned experts.  In addition to the conventional MoE loss, DyMoE introduces a block-guided regularisation loss to correctly assign experts for different data blocks. To improve the efficiency of the DyMoE model, the paper also proposes a sparse DyMoE model, where instead of fusing the predictions from all the expert networks, it only uses the top-K most relevant experts to make predictions.

**Strengths:**

The paper is easy to follow. The idea of using the MoE model to address graph incremental learning is novel and interesting. Experimental results on six graph incremental learning datasets demonstrate the effectiveness of the proposed DyMoE model and the block-guided regularisation loss. The results also indicate the DyMoE model can learn dedicated experts for different data blocks.

**Weaknesses:**

1. Theorem 1 is established under the assumption that the data follow a Gaussian mixture distribution, and this assumption should be explicitly stated in the theorem to make it more precise. Can this theorem extend to data following distributions other than the Gaussian mixture distribution?

2. The details of the data balancing training procedure (line 297-301) is not very clear from the paper. Specifically, how to select the memory set for the new data block? Does this training procedure use both $L_{cls}$ and $L_{BL}$ loss?

3. There are several missing details in the experimental setups that should be clarified:

(1) The number of nodes and edges per data block is not provided in Table 8. Moreover, the paper only introduces how to split the data into different blocks in Appendix C.2. However, it is unclear how to obtain the training, validation and test sets in each block.

(2) It is unclear which GNN model was used in the experiments, and whether the same GNN model was applied to the baseline models.

(3) Details of the evaluation metrics are missing, including how each metric is calculated.

4. The analysis of the results is inaccurate and unconvincing in some cases:

(1) In line 429, the paper states that “we can see our method significantly improves over existing baselines for both AA and AF”. However, this is inaccurate since the performance of DyMoE (57.85) is actually worse than the baseline PI-GNN (59.18) on DBLP in terms of AA.

(2) In line 430, the paper states that “We reach an average of 5.47% improvement in AA and 34.64% reduction in AF”. However, it didn’t specify which model these improvements are compared to.

(3) The results in Table 1 and Table 2 indicate that the proposed DyMoE model performs worse than baselines on the DBLP and Arxiv in terms of AA. However, the paper lacks analysis to explain this result.

(4) I also have some concerns regarding the efficiency experiments. In table 3, the paper provides the training time of DyMoE and three baselines: Finetune, ER-GNN and Retrain. In table 5, the paper only compares DyMoE and ER-GNN in terms of inference time. It is unclear why the paper only compares these three baselines instead of the more effective baseline C-GNN. Moreover, the results in table 5 show that the inference time of DyMoE is worse than ER-GNN, which indicates that the proposed DyMoE model actually cannot achieve good performance while maintaining good efficiency.

(5) In table 4, I don’t think the comparison between Full and the other variants is fair given that all the other variants are based on the sparse model. Comparing the results of Sparse and w/o DB, w/o BL, w/o Dy, the statement that “we see performance drop whenever a component is missing from the model, validating the importance of each component” is not very accurate as the performance of w/o Dy (92.09) is better than Sparse (91.57) on Reddit in terms of AA and the performance of w/o DB (83.09) is better than Sparse (82.93) on Paper100M in terms of AA.

5. The effect of the hyperparameter $K$ in the sparse model is not investigated.

6. The code is not available, which makes it difficult to reproduce the results given that a lot of experimental details are missing.

**Questions:**

1. Can Theorem 1 extend to data following distributions other than Guassian mixture distribution?
2. In the efficiency analysis, why not report the training and inference runtime of more effective baselines such as C-GNN?
3. The proposed DyMoE learns a new expert when a new data block arrives. This requires prior knowledge of which data belong to different blocks and a sufficient amount of data in each block to effectively learn a new expert. This assumption may present challenges in real-world incremental learning applications, where data blocks are not predefined or large enough for expert learning.

---

> ### Author Response · Authors · 2024-11-20
> **Author response 1**
>
> We sincerely thank you for your detailed comments to help us improve the paper, and we would like to address some of your concerns here.
>
> > Theorem 1 is established under the assumption that the data follow a Gaussian mixture distribution, and this assumption should be explicitly stated in the theorem to make it more precise. Can this theorem extend to data following distributions other than the Gaussian mixture distribution?
>
> We agree with you and modify the theorem to be more accurate in our revision. Meanwhile, theorem 1 consists of two parts, in the first part we show that DyMoE is at least as powerful as PI in arbitrary distribution. In the second part, we show that DyMoE is more powerful than PI under the Gaussian Mixture assumption. These two parts combined show that DyMoE is generally more favorable than PI. In terms of the theorem’s generalizability, we observe that in our proof, the only Gaussian property it relies on is the Gaussian Tail bound. In the proof, we can replace the bound with the definition of a sub-gaussian probability, and the theorem will still hold (sub-gaussian already includes a large class of distributions, such as symmetric Bernoulli, symmetric triangular).
>
> > The details of the data balancing training procedure (line 297-301) is not very clear from the paper.  Specifically, how to select the memory set for the new data block?
>
> Following many previous works [1,2], the memory node set is selected as top-K nodes whose embedding is closest to the class embedding. The K for each class is determined by the population. Let $d$ be the overall memory budget for a data block $B$, for one class $c$, $K = \frac{|B_c|}{|B|}*d$.
>
> We perform data balancing training after training the new data block $B$. We first obtain its memory set $S_B$, and take the union of the memory set with all previous memory sets to get the overall memory set $S$. Finally, we train the model on S for a small number of epochs (e.g. 5 epochs) with classification loss and block-guided loss.
>
> We will include a detailed description in the revision.
>
> > The number of nodes and edges per data block is not provided in Table 8. Moreover, the paper only introduces how to split the data into different blocks in Appendix C.2. However, it is unclear how to obtain the training, validation and test sets in each block.
>
> For CoraFull and Reddit, we use the split provided by the DGL package, for ArxivIIL, we use the 0.6/0.2/0.2 train/val/test split within each data block, for ArxivCIL, we use the original split provided by the OGB package. For DBLP and paper100M, we use the split provided by PI-GNN.
>
> For the specific number of nodes and edges, because the datasets have up to 49 data blocks, we present them as line plots in the updated pdf.
>
> > It is unclear which GNN model was used in the experiments, and whether the same GNN model was applied to the baseline models.
>
> We strictly follow the GNN architecture described in line 323 of the paper, which is a GIN with the proposed DyMoE module. For all other baselines, we also use GIN, if applicable.
>
> > Details of the evaluation metrics are missing, including how each metric is calculated.
>
> The evaluation metrics are average accuracy and average precision, which are described in the preliminary section in Equation 1.
>
> > In line 429, the paper states that “we can see our method significantly improves over existing baselines for both AA and AF”. However, this is inaccurate since the performance of DyMoE (57.85) is actually worse than the baseline PI-GNN (59.18) on DBLP in terms of AA. AND In line 430, the paper states that “We reach an average of 5.47% improvement in AA and 34.64% reduction in AF”. However, it didn’t specify which model these improvements are compared to.
>
> We will revise the experimental section to more accurately analyze the results. Here, we are referring to the overall performance. Because none of the baselines achieves unanimously the best performance on all datasets, the average performance improvement is computed over the best baselines for each dataset to make it a more strict and challenging evaluation for DyMoE. Specifically, we compare DyMoE with C-GNN on Corafull, arxiv, and Reddit for AA, because C-GNN is the best baseline on these datasets, and we compare with PI-GNN on DBLP as PI-GNN is the best baseline on DBLP. We then take the average change as the overall performance improvement.  Under this protocol, we achieve a 5.47% average improvement in AA and a 34.64% average reduction in AF, which show that overall DyMoE is more optimal.

---

> ### Author Response · Authors · 2024-11-20
> **Author Response 2**
>
> > The results in Table 1 and Table 2 indicate that the proposed DyMoE model performs worse than baselines on the DBLP and Arxiv in terms of AA. However, the paper lacks analysis to explain this result.
>
> Our method still outperforms other baseline on the Arxiv-Class incremental learning scenario. For the instance incremental scenario, we note that even the most basic baseline (Online GNN) is performing very well, showing that Arxiv Instance Incremental does not carry significant distribution shift that DyMoE is good at capturing, and DyMoE uses more parameters, which makes the model prone to overfitting more.
> 	For the DBLP dataset, we observe a trade-off between average accuracy and average forget. (Methods with higher accuracy get worse forget) For this dataset, it is difficult to learn new knowledge, without compromising some previously acquired knowledge. Hence we observe that DyMoE is good at maintaining a low forgetting while sacrificing some accuracy.
>
> > Efficiency Concern
>
> For the training time, we pick these baselines for running time comparison because they are the most representative baselines. In particular, C-GNN has the same running time as ER-GNN when their memory set sizes are the same (which is the case in this paper). Hence, we only provided the running time of the most representative ones.
>
> For the inference time, because finetune, ER-GNN and Retrain have the same architecture, they will have the same running time as well, which is why we provided the results for ER-GNN.
>
> In the appendix, we conducted experiments when we increased the sizes of baseline models and found that their running time increased with minimal or even negative performance improvement. We present results when we make DyMoE’s parameter sizes the same as other baseline models in general response.
>
> We can see that, when DyMoE and baselines have the similar number of active parameters, the inference time is similar, and DyMoE can still achieve promising results.
>
> > Concerns about ablation study.
>
> We appreciate your detailed look into the paper, we will carefully revise this section to accurately reflect the experimental results. Meanwhile, we still observe that without data balancing, the overall average accuracy dropped by 1.8%, showing the necessity to inform the gating vectors the actual distribution of the data blocks. Without block-guided loss, the overall AA dropped by 3.8%, validating block-guided loss is necessary to train specialized experts. If the model is trained with all experts initialized at the first data block, the performance dropped by 2.2%, as this essentially degrades to a regular MoE model, which does not have a mechanism to prevent forgetting.
>
> For Paper100M, we observe that the number of data samples per block increases, making more recent blocks have a higher weight in the overall accuracy. In this case, accurately representing the full distribution through DB is not as critical. For reddit, we observe that it is much denser then other datasets, requiring more parameters to learn the graph, and with more initial parameters, the w/o Dy model can model the first few data blocks better. Because reddit only has 8 data blocks, the benefit of DyMoE in choosing correct experts is not apparent.
>
> > Effectiveness of $K$
>
> We additionally present results when we increase K. We can see that even with just one active expert, DyMoE is performing well, showing the effectiveness of the specialized experts. We also observe that K has a different impact on different data. Specifically, the benefit of more active experts saturates quite early for the arxiv dataset, while the reddit dataset continuously benefits from more active experts, which is intuitive as the gap between the full DyMoE and sparse DyMoE(K=3) is larger.
>
> |     | Arxiv | Reddit |
> |-----|-------|--------|
> | k=1 | 66.53 | 88.46  |
> | k=3 | 67.25 | 91.57  |
> | k=5 | 68.14 | 91.98  |
> | k=7 | 68.09 | 92.63  |
>
> > Code availability
>
> We will release the code after the final decision.

---

> ### Author Response · Authors · 2024-11-20
> **Author Response 3**
>
> > The proposed DyMoE learns a new expert when a new data block arrives. This requires prior ...
>
> Compared to other methods, the block-guided loss, which directly correlates samples to their corresponding experts, is designed to better distribute samples to the most relevant experts during inference.
>
> In case of very small block sizes, it will not shift the entire distribution, and we might consider simply using the existing model for prediction or combining the new data into the latest block and retrain the last export. If we fine-tune on the small data block, most existing methods will suffer from overfitting due to the small sample size. On the other hand, updating the model with a small data block is not economical and favorable in production. People usually control the intervals between model refreshes to balance the model performance and the cost,  and they typically update their models at regular intervals, such as daily or weekly, to accumulate sufficient numbers of samples.
>
> However, we agree with you that the uneven block sizes present unique challenges to continual learning, and is a valuable setting to explore and present experimental results where the sizes of data blocks in a data stream are uneven.
>
> [1] Zhou, Fan, and Chengtai Cao. "Overcoming catastrophic forgetting in graph neural networks with experience replay." Proceedings of the AAAI Conference on Artificial Intelligence. Vol. 35. No. 5. 2021.
>
> [2] Kim, Seoyoon, Seongjun Yun, and Jaewoo Kang. "DyGRAIN: An Incremental Learning Framework for Dynamic Graphs." IJCAI. 2022.

---

### Official Review · Reviewer_8EcB · 2024-11-04

**Soundness:** 4
**Presentation:** 3
**Contribution:** 4
**Rating:** 6
**Confidence:** 4

**Summary:**

This paper proposes a Dynamic Mixture-of-Experts (DyMoE) approach for graph incremental learning, utilizing specialized expert networks for incoming data blocks. It introduces a customized regularization loss to help existing experts retain performance on old tasks while supporting new learning. Additionally, a sparse MoE model is developed to reduce computational costs by using only the most relevant experts for predictions.

**Strengths:**

1.	This paper introduces a Dynamic Mixture-of-Expert (DyMoE) module with separate experts for each data block, allowing dynamic relevance-based information synthesis.
2.	This paper proposes a block-guided loss function to minimize negative interference among experts, reducing catastrophic forgetting.
3.	This paper integrates the DyMoE module into GNN layers to effectively handle data shifts in continual graph learning.
4.	This paper develops a sparse DyMoE variant that focuses on the most relevant experts, enhancing efficiency while maintaining accuracy.

**Weaknesses:**

1.	In real-world applications, how do the dynamic changes in graph structures and data blocks affect the model's performance? Have you considered the impact of data noise and outliers on the results?
2.	Experiments:
1)	The MoE structure increases the number of parameters in the model, thereby enhancing its capability. In contrast, the parameter count of the baseline model is not specifically mentioned. Does this represent an unfair comparison?
2)	This paper mentions "with minimal computation increase," but it seems insufficient experimental or theoretical evidence is provided. Could there be a comparison of throughput?
3)	The paper claims that this method can handle topological and contextual changes, but how is this topological change quantified and assessed? Understanding the extent and nature of these changes is crucial for evaluating the effectiveness of the proposed approach. There are no experiments to address this problem.

**Questions:**

1. How do the dynamic changes in graph structures and data blocks affect the model's performance? Have you considered the impact of data noise and outliers on the results?

2. Enhance experiments and analysis.

---

> ### Author Response · Authors · 2024-11-20
>
> We greatly appreciate your positive comments and critical feedback on our work, we address your concerns as follows.
>
> > In real-world applications, how do the dynamic changes in graph structures and data blocks affect the model's performance? Have you considered the impact of data noise and outliers on the results?
>
> That’s a great question. It is indeed more challenging when the data are noisy and dynamic. We can compare the performance of the Pretrain baseline (we only train the model using the first data block and run inference on the following data blocks with this model) with other existing baselines, and we can see a large performance gap because the distribution of the data significantly shifted. More interestingly, we observe that when the graph changes more rapidly, such as in arxiv where the volume of publications significantly increases over the past few decades, the gap is large, and the gap is smaller when the graph is more consistent, such as in Elliptic where the data only span roughly 2 years. To cope with this challenge, we propose block-guided loss to stabilize training. For example, when an outlier falls in the distribution boundary of two data blocks, it would be difficult for the MoE model alone to determine the most relevant expert. However, in the case of continual learning, every data point is accompanied by a block index, and hence we can use that information to provide direct supervision to reduce the risk of assigning wrong experts to noisy/outlier data points.
>
> > The MoE structure increases the number of parameters in the model, thereby enhancing its capability. In contrast, the parameter count of the baseline model is not specifically mentioned. Does this represent an unfair comparison?
>
> It is indeed important to evaluate the models under the same parameter budget, and we present results in general response and show that the fixed-sized parameters model experience performance degradation when they are given the same parameter budget as DyMoE, mostly due to the overfitting.
>
> > This paper mentions "with minimal computation increase," but it seems insufficient experimental or theoretical evidence is provided. Could there be a comparison of throughput?
>
> Compared to the complexity/running time lower bound finetune on the cora-full dataset, we observe that the training time increases by 7%, whereas we observe a 109% performance increase. Compared to the ER method, the training time increase is 4%, whereas the performance increase is 14.4%. This shows that DyMoE achieves higher performance in an efficient way.
>
> From a theoretical standpoint, the complexity of Training DyMoE is $O(nkT)$, where $n$ is the number of samples, $k$ is the number of active experts, and $T$ is the complexity of a single expert,  the complexity of online GNN is $O(nT)$, because k is usually a small constant, the running time of DyMoE is comparable to the lower bound online GNN. On the contrary, the retraining method has a complexity of $O(ntT)$, where $t$ is the number of data blocks, and $t$ is usually large ($t > 10$), leading to higher computational costs.
>
> >The paper claims that this method can handle topological and contextual changes, but how is this topological change quantified and assessed? Understanding the extent and nature of these changes is crucial for evaluating the effectiveness of the proposed approach. There are no experiments to address this problem.
>
> Note that there is no golden rule/single metric to determine and quantify the distribution shift. Moreover, not all structural changes impact the labeling process. Hence, we did not include a distribution shift in the submission. However, we agree it will be more tangible to include some structural changes across data blocks to understand the problem and proposed method, and hence provide the change of several graph properties to understand the evolution of the data blocks. Because most datasets have over 10 data blocks, we present the overall standard deviation and mean of the graph properties for clarity. Specifically, we compute two important graph properties, the average number of triangles per node and the graph density, for each data block, and we compute the standard deviation and mean of these properties across data blocks, to show the change in graph structure.
>
> |          | Ave # triangles (Mean) | Ave # triangles (std) | Density (Mean) | Density(std) |
> |----------|------------------------|-----------------------|----------------|--------------|
> | arxiv    | 14.45                  | 12.95                 | 6.33E-05       | 2.54E-05     |
> | coraFull | 5.69                   | 1.21                  | 0.000427       | 0.000469     |
>
> From the results we can see that both properties have large standard deviations, validating the necessity of accounting for the structural shift.

---

### Author Response · Authors · 2024-11-20
**General Response**

We would like to thank all reviewers for their constructive feedback and for acknowledging some of our contributions. In particular,
- The identified problem is important and interesting. (Reviewer Ejrw, 6WFG)
- The proposed method “effectively handles data shifts” through block-guided loss and DyMoE model. (Reviewer 8EcB, VebG)
- The method is empirically verified with experiments and shows strong performance. The paper also carefully examined the design to show that DyMoE indeed results in specialized experts to reduce forgetting. (Reviewer 8EcB, VebG, Ejrw)

The reviewers also express some concerns about the proposed methods, here we address some of the common concerns and leave the rest to individual responses to the reviewers.

> The continual learning problem involves trade-off among inference time, parameter size and performance, how does DyMoE respond to this?

Here we present experimental results comparing DyMoE with other baselines varying active parameter sizes.

|            | Active Params | Reddit | Time  | Arxiv-CIL | Time  |
|------------|---------------|--------|-------|-----------|-------|
| ER-GNN     | 26M           | 81.35  | 18.41 | 57.09     | 11.07 |
| C-GNN      | 26M           | 86.75  | 18.37 | 63.65     | 10.94 |
| DyMoE(k=3) | 28M           | 90.06  | 17.61 | 66.09     | 11.02 |
| ER-GNN     | 80M           | 80.69  | 19.89 | 59.44     | 14.61 |
| C-GNN      | 80M           | 86.3   | 20.01 | 62.18     | 14.78 |
| DyMoE(k=3) | 77M           | 91.57  | 20.55 | 67.25     | 13.65 |

We can see that when models have similar parameter sizes, the inference time of DyMoE is comparable to other baselines, and still achieves better performance.

> Comparing DyMoE with more recent baselines.

We compared DyMoE with two more recent baselines RLC-CN and SSRM-GIN.

|          | Paper100 AA | Paper100 AF | CoraFull AA | CoraFull AF | Reddit AA | Reddit AF |
|----------|-------------|-------------|-------------|-------------|-----------|-----------|
| RLC-CN   | 76.53       | -4.61       | 72.94       | -9.8        | 82.39     | -5.52     |
| SSRM-GIN | 81.6        | -3.8        | 82.49       | -6.37       | 88.45     | -6.14     |
| DyMoE    | 82.93       | -3.31       | 81.33       | -5.69       | 91.57     | -3.46     |

We can see that DyMoE’s performance is still strong compared to more recent baselines. While SSRM-GIN outperforms DyMoE on CoraFull AA, we notice that SSRM is regularization approaching, optimizing the model to fight structural shift through an auxiliary loss. We can potentially incorporate this into DyMoE to further improve performance.


[1] ''Reinforced continual learning for graphs,'' in Proceedings of the 31st ACM International Conference on Information & Knowledge Management, 2022, pp. 1666–1674.

[2] "Towards robust graph incremental learning on evolving graphs." International Conference on Machine Learning. PMLR, 2023.

---

### Meta-Review · Area_Chair_1gvw · 2024-12-23

**Metareview:**

This paper looks at the graph incremental learning problem which is basically the continual learning problem where each task involves learning a graph neural network (GNN). The paper presents a dynamic mixture of experts (MoE) approach in which a dynamic MoE GNN layer adds new expert networks dedicated to model the incoming data for a new task.

The paper's basic idea of using MoE is interesting; however, it is to be noted that MoE has been used in several prior works in continual learning although not for the setting where each task is a GNN.

The reviewers expressed several concerns, some of which include: (1) issues regarding the motivation and validation, and randomization mechanism used in expert section  (Reviewer Ejrw), (2) missing SOTA baselines such as MSCGL, RLC-CN, SEM, and UGCL  (Reviewer Ejrw), (3) lack of discussion/insights about the challenges in applying MoE for incremental GNN (Reviewer 6WFG), (4) lack of discussion around model size for large data stream and how the proposed approach compared with alternatives such as those that rely on parameter/information sharing among tasks  (Reviewer 6WFG).

The authors during rebuttal/discussion tried to address some of these concerns but the concerns still lingered. In the end, no one championed the paper for acceptance.

Based on the reviews, the authors' response, the discussion, and my own reading of the paper, I largely agree with the concerns raised by the reviewers. The authors should take into account the feedback to look for ways to improve the work and consider submitting at another venue.

**Additional Comments On Reviewer Discussion:**

The authors' rebuttal was considered and discussed.

Reviewer Ejrw expressed concerns regarding the motivation and validation of the method, and about the selection of the last expert. The authors responded to the concern but the reviewer maintained that the reasoning behind the method's motivation and validation appear to be "circular". The reviewer's concern regarding the randomization mechanism in expert selection also remained unresolved. Due to these reasons, Reviewer Ejrw maintained the original score.

Reviewer 6WFG also expressed several concerns, such as differences from prior works in graph continual learning. The authors responded to these in detail but in the follow-up discussions, these concerns lingered.

No other reviewer championed the paper.

The decision was made after factoring in all these points.

---

### Decision · Program_Chairs · 2025-01-22

Reject